# Exome sequencing and analysis of 454,787 UK Biobank participants

Joshua D. Backman[1], Alexander H. Li[1], Anthony Marcketta[1], Dylan Sun[1], Joelle Mbatchou[1], Michael D. Kessler[1], Christian Benner[1], Daren Liu[1], Adam E. Locke[1], Suganthi Balasubramanian[1], Ashish Yadav[1], Nilanjana Banerjee[1], Christopher E. Gillies[1], Amy Damask[1], Simon Liu[1], Xiaodong Bai[1], Alicia Hawes[1], Evan Maxwell[1], Lauren Gurski[1], Kyoko Watanabe[1], Jack A. Kosmicki[1], Veera Rajagopal[1], Jason Mighty[1], Regeneron Genetics Center*, DiscovEHR*, Marcus Jones[1], Lyndon Mitnaul[1], Eli Stahl[1], Giovanni Coppola[1], Eric Jorgenson[1], Lukas Habegger[1], William J. Salerno[1], Alan R. Shuldiner[1], Luca A. Lotta[1], John D. Overton[1], Michael N. Cantor[1], Jeffrey G. Reid[1], George Yancopoulos[1], Hyun M. Kang[1], Jonathan Marchini[1,2], Aris Baras[1,2 ✉], Gonçalo R. Abecasis[1,2 ✉] & Manuel A. R. Ferreira[1,2 ✉]

A major goal in human genetics is to use natural variation to understand the phenotypic consequences of altering each protein-coding gene in the genome. Here we used exome sequencing[1] to explore protein-altering variants and their consequences in 454,787 participants in the UK Biobank study[2]. We identified 12 million coding variants, including around 1 million loss-of-function and around 1.8 million deleterious missense variants. When these were tested for association with 3,994 health-related traits, we found 564 genes with trait associations at $P \leq 2.18 \times 10^{-11}$. Rare variant associations were enriched in loci from genome-wide association studies (GWAS), but most (91%) were independent of common variant signals. We discovered several risk-increasing associations with traits related to liver disease, eye disease and cancer, among others, as well as risk-lowering associations for hypertension (*SLC9A3R2*), diabetes (*MAP3K15*, *FAM234A*) and asthma (*SLC27A3*). Six genes were associated with brain imaging phenotypes, including two involved in neural development (*GBE1*, *PLD1*). Of the signals available and powered for replication in an independent cohort, 81% were confirmed; furthermore, association signals were generally consistent across individuals of European, Asian and African ancestry. We illustrate the ability of exome sequencing to identify gene–trait associations, elucidate gene function and pinpoint effector genes that underlie GWAS signals at scale.

A major goal in human genetics is to use natural variation to understand the consequences of altering each protein-coding gene in the genome. Towards that goal, the UK Biobank (UKB) Exome Sequencing Consortium[1] sequenced the exomes of 454,787 UKB participants (Supplementary Table 1), with 95.8% of targeted bases covered at a depth of 20× or greater, as previously described[1,3]. We identified 12.3 million variants in 39 million base pairs across the coding regions of 18,893 genes (Table 1), of which 99.6% were rare variants (minor allele frequency (MAF) < 1% across all ancestries). This catalogue exceeds by about 1.3-fold the coding variation contained in the combined TOPMed[4] and gnomAD[5] datasets (9.5 million autosomal variants), and by about 8-fold the coding variation accessible in the UKB through imputation (1.6 million autosomal variants with info score > 0.3; Supplementary Table 2). Among the variants identified were 3,457,173 (median of 10,273 per individual) synonymous, 7,878,586 (9,292 per individual) missense and 915,289 (214 per individual) putative loss-of-function (pLOF) variants (Table 1), of which about half were observed only once

in this dataset (singleton variants; Supplementary Fig. 1). About 23% (1,789,828) of missense variants were predicted to be deleterious by 5 prediction algorithms (see Methods; henceforth 'deleterious missense variants'). This unique catalogue of coding variation, combined with the large sample size and thousands of available phenotypes, provides a unique opportunity to assess gene function at a large scale.

## Association studies of rare variants

GWAS often do not elucidate gene function per se because (i) most protein-coding variants are not accessible through imputation (Supplementary Table 3); and (ii) it is not straightforward to identify the specific genes and mechanisms that underlie associations with common non-coding variants[6]. To illustrate the potential to elucidate gene function through analysis of whole-exome sequencing (WES) data, we tested the association between rare pLOF and deleterious missense variants and 3,994 health-related traits measured in participants in the UKB

[1]Regeneron Genetics Center, Tarrytown, NY, USA. [2]These authors jointly supervised this work: Jonathan Marchini, Aris Baras, Gonçalo R. Abecasis, Manuel A. Ferreira. *Lists of authors and their affiliations appear in the Supplementary Information. ✉e-mail: goncalo.abecasis@regeneron.com; manuel.ferreira@regeneron.com

**Table 1 | Number of coding variants discovered in exome sequencing data from 454,787 participants in the UK Biobank**

| Variant category | No. of variants (% with MAC = 1) | Median number of variants per participant (IQR) |
|---|---|---|
| Coding regions[a] | 12,326,144 (46.86) | 19,895 (247) |
| **Predicted function** | | |
| In-frame indels | 75,096 (40.33) | 115 (11) |
| Synonymous | 3,457,173 (43.12) | 10,273 (141) |
| Missense | 7,878,586 (47.28) | 9,292 (143) |
| Likely benign | 1,532,129 (44.11) | 6,561 (104) |
| Possibly deleterious | 4,556,629 (47.23) | 2,610 (70) |
| Likely deleterious | 1,789,828 (50.1) | 121 (16) |
| pLOF (any transcript) | 915,289 (57.88) | 214 (16) |
| Start lost | 26,453 (47.94) | 13 (4) |
| Stop gained | 279,913 (54.02) | 52 (8) |
| Stop lost | 12,843 (56.51) | 6 (3) |
| Splice donor | 104,328 (58.67) | 17 (5) |
| Frameshift | 405,669 (60.41) | 90 (10) |
| Splice acceptor | 86,083 (60.79) | 20 (5) |

[a]Includes all coding variants: synonymous, in-frame indels, missense and pLOF variants. MAC, minor allele count; IQR, interquartile range.

study (Supplementary Data 1). This included 3,702 binary traits with at least 100 cases and 292 quantitative traits from a variety of domains, including anthropometry, biochemistry and haematology (Supplementary Table 4). About half of the binary traits were uncommon, with a population prevalence between 0.1% and 1% (Supplementary Fig. 2). Association analyses were performed using the whole-genome regression approach implemented in REGENIE[7], which accounts for relatedness, population structure and polygenicity and uses a fast, approximate Firth regression approach for binary outcomes. Variants were tested individually and on aggregate, through gene burden tests that group protein-altering variants within each gene.

We first analysed WES data from individuals of European ancestry ($n$ = 430,998; around 95% of the total sample size), focusing on pLOF (including stop-gain, frameshift, stop-lost, start-lost and essential splice variants) and deleterious missense variants with a MAF of up to 1%. We tested for association between each trait and individual variants in 18,811 genes, as well as with aggregations of variants in each gene, considering either pLOF or pLOF and deleterious missense variants jointly. Overall, we performed a total of around 2.3 billion association tests (Supplementary Table 5), with no evidence for a substantial effect of population structure or unmodelled relatedness on the results (Supplementary Figs. 3, 4). We found 8,865 significant associations—involving 564 genes, 492 traits and 2,283 gene–trait pairs (Extended Data Fig. 1)—at $P \le 2.18 \times 10^{-11}$, which corresponds to a Bonferroni correction for multiple testing (that is, $P \le 0.05/2.3$ billion tests; at this threshold, <0.05 association signals expected by chance across the full result set). As we show later, 8,059 (91%) of these associations could not be explained by linkage disequilibrium (LD) with nearby common variants and, furthermore, 81% of associations available and powered for replication were confirmed in an independent but smaller cohort of $n$ = 133,370 individuals (DiscovEHR cohort). All 8,865 associations are provided in Supplementary Data 2, as well as two non-redundant sets that were obtained by retaining only the most significant signals: (i) per gene–trait pair (2,283 signals; filtered view in Supplementary Data 2); or (ii) per gene (564 signals; Supplementary Table 6). Of the 564 lead gene associations, 415 were due to a burden signal (which typically aggregated single-nucleotide polymorphisms (SNPs) and indels) and 149 were due to an individual rare variant. Of these 149, 20 represented

association with an indel variant and 129 represented association with a single-nucleotide variant (SNV) (Supplementary Table 6). Gene targets of drugs approved by the Food and Drug Administration were 3.6-fold more common among the associated genes (36 of 564, or 6.4%; Supplementary Table 6) than in the remaining genes (345 of 18,317, or 1.9%; Fisher's exact test $P = 1.7 \times 10^{-9}$).

The large number of associations identified provides an opportunity to understand the phenotypic consequences of protein-altering variation in humans and identify therapeutic targets. As it is not possible to exhaustively describe all novel gene associations, we instead highlight examples selected from four broad groups of variants: (i) singleton variants; (ii) risk lowering variants; (iii) variants with a beneficial effect on a quantitative trait; and (iv) variants likely to be of somatic origin. These groupings illustrate the value of the UKB exome resource and the potential of our data to power further discovery and analyses.

## Associations with singleton variants

We first focused on 69 signals that were discovered when considering a burden of singleton variants, which represent the rarest class of variation and remain well beyond the reach of genotyping arrays and imputation using existing reference panels. Association of a phenotype with the burden of singletons in a gene represents one of the most compelling ways for human genetics to implicate a gene in disease[8]. Each of the 69 genes was associated with an average of 5.7 (mostly correlated) traits, resulting in a total of 393 associations (4.4% of the total; Supplementary Data 2). To our knowledge, 15 of these 69 gene associations have not been previously described (Extended Data Table 1), of which we highlight 2. First, carriers of singleton pLOF variants in the chromatin remodeller *EP400* had lower hand grip strength (96 carriers; effect = −0.55 s.d. units, 95% confidence interval (CI) −0.68 to −0.42, $P = 8 \times 10^{-16}$), consistent with findings from knock-out mice, which also present peripheral neuropathy and severe hypomyelination of the central nervous system[9]. Second, singleton pLOF variants in *RRBP1*, which encodes an endoplasmic reticulum membrane protein, were associated with lower levels of apolipoprotein B (92 carriers; effect = −0.83 s.d. units, 95% CI −1.0 to −0.64, $P = 3 \times 10^{-18}$), as well as similar reductions in the levels of low-density lipoprotein and total cholesterol. Consistent with this, silencing of *Rrbp1* in mice altered hepatic lipid homeostasis, resulting in reduced biogenesis of very-low-density lipoprotein[10].

## Protective associations with disease outcomes

A major impetus to perform association analyses with rare variants is the identification of genes for which loss-of-function variants are associated with lower disease risk, as these may represent attractive targets for blocking antibodies or other inhibitory modalities. However, power to identify protective associations with rare variants at $P \le 2.18 \times 10^{-11}$ was low (Extended Data Fig. 2). Consistent with this, we found only five genes associated with a lower risk of disease outcomes at $P \le 2.18 \times 10^{-11}$, all previously reported: *PCSK9*, *APOB* and *APOC3* and protection from hyperlipidaemia; *ABCG5* and cholelithiasis; and *IL33* and allergic diseases (Supplementary Table 7).

Of note, however, an additional 11 protective associations were observed at a more liberal significance threshold of $P \le 10^{-7}$, including 6 previously reported (involving *ANGPTL3*, *IFIH1*, *DBH*, *PDE3B*, *SLC22A12* and *ZNF229*) and 4 that are potentially novel and remain highly associated after accounting for common variant signals (Supplementary Table 7). The first was between *SLC9A3R2* and lower risk of hypertension (5,873 carriers; odds ratio (OR) = 0.81, 95% CI 0.76 to 0.87, $P = 2.2 \times 10^{-10}$). There were also strong associations when systolic blood pressure (SBP; effect = −1.85 mmHg, 95% CI −2.22 to −1.48, $P = 2.0 \times 10^{-19}$) and diastolic blood pressure (DBP; effect = −1.01 mmHg, 95% CI −1.31 to −0.80, $P = 4.8 \times 10^{-18}$; Supplementary Data 2) were analysed as quantitative traits, with the SBP association replicating in the DiscovEHR cohort ($P = 2.6 \times 10^{-4}$; Supplementary Table 6). *SLC9A3R2* encodes NHERF-2, a kidney-expressed scaffolding protein that is functionally linked

to sodium absorption through interaction with sodium/hydrogen exchanger 3[11]. An association with a low-frequency missense variant in *SLC9A3R2* (rs139491786, Arg171Trp, MAF = 0.7%) was previously identified in a GWAS of blood pressure[12], but the signal was attributed to a nearby variant in *PKD1* (rs140869992, Arg2200Cys). We show that a burden of rare pLOF and deleterious missense variants in *SLC9A3R2*, as well as Arg171Trp, remain highly associated with SBP, DBP and hypertension after conditioning on Arg2200Cys in *PKD1* (Supplementary Table 8). Overall, the signal is consistent with the well-established role of sodium balance in regulating blood pressure and suggests that blocking *SLC9A3R2* could provide a means for managing blood pressure. Functional and clinical studies that evaluate this possibility are warranted.

The second novel association was between lower risk of childhood asthma and a burden of rare pLOF and deleterious missense variants in *SLC27A3* (3,787 carriers; OR = 0.65, 95% CI 0.55 to 0.76, $P = 8.2 \times 10^{-8}$), which was supported by the following additional observations. First, a burden of rare pLOF and deleterious missense variants was also associated with lower blood eosinophil counts (5,227 carriers; effect = −0.045 s.d. units, 95% CI −0.070 to −0.020, $P = 4.4 \times 10^{-4}$), a cell type with critical effector functions in allergic asthma. Second, there were consistent protective associations in the DiscovEHR cohort with both asthma (1,354 carriers; OR = 0.87, 95% CI 0.75 to 1.01, $P = 0.060$) and eosinophil counts (1,755 carriers; effect = −0.052 s.d. units, 95% CI −0.096 to −0.008, $P = 0.021$). *SLC27A3* encodes an acyl-CoA synthetase that activates long-chain fatty acids[13], is most highly expressed in artery, adipose and lung tissue[14] and is upregulated in lung cancer[15].

The third novel association was between a missense variant in *PIEZO1* (rs61745086, Pro2510Leu, MAF = 0.98%) and reduced risk of varicose veins (7,454 carriers; OR = 0.69, 95% CI 0.61 to 0.79, $P = 2.61 \times 10^{-8}$). *PIEZO1* encodes a mechanosensitive cation channel with a key role in venous and lymphatic valve formation[16]. We had previously shown that rare pLOFs in this gene increase the risk of asymptomatic varicose veins of lower extremities by 4.9-fold (162 carriers; 95% CI 2.8 to 8.6, $P = 3.2 \times 10^{-8}$) in the first 49,960 exomes from the UKB[3], an association that is now estimated at 2-fold with around 8 times more data (1,355 carriers; OR = 2.08, 95% CI 1.62 to 2.67, $P = 7.4 \times 10^{-9}$). The new protective association with Pro2510Leu, which replicated in the DiscovEHR cohort (2,243 carriers; OR = 0.66, 95% CI 0.47 to 0.93, $P = 0.017$), suggests that this missense variant probably has a gain-of-function effect. This is important because it suggests that activation of *PIEZO1* may provide a therapeutic pathway for a common condition with no available pharmacological interventions.

Finally, the fourth novel association was between *MAP3K15* and protection from type-2 diabetes, which is discussed in greater detail below. Among these four novel protective associations, only two (with *SLC9A3R2* and *PIEZO1*) were observed at $P < 10^{-7}$ when analysing TOPMed imputed data (Supplementary Tables 9, 10).

### Protective associations with quantitative traits

The low yield of protective associations with disease traits contrasts with that observed for disease-relevant quantitative traits, such as body mass index, which often provide greater power for genetic studies. Specifically, we found 131 genes for which the direction of effect on a quantitative trait was consistent with a beneficial effect on disease risk (Supplementary Table 11). For example, we found low-frequency protein-altering variants in *ASGR1* associated with lower apolipoprotein B levels (759 carriers; effect = −0.29 s.d. units, 95% CI −0.35 to −0.22, $P = 6.5 \times 10^{-18}$). *ASGR1* haploinsufficiency was previously reported to reduce risk of cardiovascular disease[17], an observation that supported the clinical development of an anti-ASGR1 monoclonal antibody as a lipid-lowering therapeutic agent[18].

As another example, we found an association between lower serum glucose levels and pLOF variants in *FAM234A* (2,439 carriers; effect = −0.14 s.d. units, 95% CI −0.18 to −0.099, $P = 2.0 \times 10^{-12}$), which was independent of associations with common variants (Supplementary Table 11, Supplementary Fig. 5). There was a consistent association in the DiscovEHR cohort with fasting glucose levels (1,132 carriers; effect = −0.046 s.d. units, 95% CI −0.099 to 0.007, $P = 0.09$), albeit not statistically significant. Of note, a common intronic variant in *FAM234A* was previously reported to associate with a lower risk of type-2 diabetes (rs9940149:A, MAF = 18%, OR = 0.95) and to co-localize with a regulatory variant that lowers the expression of *FAM234A* in multiple tissues[19]. Consistent with this, we found that rare pLOFs in *FAM234A* were associated with a 36% reduction in the risk of self-reported diabetes (2,104 carriers; OR = 0.64, 95% CI 0.52 to 0.80, $P = 10^{-4}$). Collectively, results from both rare and common variants implicate *FAM234A*, a gene of unknown function, in the aetiology of diabetes.

We then determined whether there were other examples of genes with both a favourable effect on a quantitative trait and a protective (even if sub-threshold) association with a relevant disease, as observed for *FAM234A*. To this end, for 131 association signals with a quantitative trait, we estimated the genetic correlation ($r_g$) between the trait and all diseases tested, and then selected the disease with the most significant $r_g$. We only considered diseases for which the $r_g$ was significant after correcting for multiple testing, if any. For example, eosinophil count was matched to asthma ($r_g = 0.37$), and intra-ocular pressure was matched to glaucoma ($r_g = 0.66$); in total, we found a matching disease for 129 trait associations (Supplementary Table 12). Using this approach, we found that 13 genes had a protective association with a genetically correlated disease that was significant after correcting for multiple testing ($P < 0.05/129$ tests = $3.8 \times 10^{-4}$; Extended Data Fig. 3). Of these, we highlight the association between a burden of protein-altering variants in *MAP3K15* and both lower levels of haemoglobin A1c (7,551 carriers; effect = −0.085 s.d. units, 95% CI −0.100 to −0.073, $P = 7.8 \times 10^{-30}$), lower serum glucose (6,885 carriers; effect = −0.090 s.d. units, 95% CI −0.110 to −0.073, $P = 1.7 \times 10^{-25}$) and protection from type-2 diabetes (7,085 carriers; OR = 0.85, 95% CI 0.79 to 0.91, $P = 2.8 \times 10^{-6}$). Furthermore, there was supporting evidence in the DiscovEHR cohort for all three phenotypes: haemoglobin A1c (1,304 carriers; effect = −0.040 s.d. units, 95% CI −0.079 to −0.002, $P = 0.038$), glucose (1,754 carriers; effect = −0.097 s.d. units, 95% CI −0.130 to −0.064, $P = 1.3 \times 10^{-8}$) and type-2 diabetes (2,455 carriers; OR = 0.91, 95% CI 0.84 to 0.98, $P = 0.018$). *MAP3K15* encodes a ubiquitously expressed, mitogen-activated protein kinase involved in apoptotic cell death[20], and has not to our knowledge previously been implicated in type-2 diabetes.

### Associations with somatic mutations

Among the 492 traits with at least 1 significant rare variant association, 20 were noteworthy because they involved 2 or more genes with rare variant signals but no common variant signals from GWAS (Extended Data Fig. 4a, b). Notably, for 7 of these 20 traits—myeloid leukaemia (7 genes; Supplementary Fig. 6), sepsis (4 genes) and 5 additional blood related traits—the majority of associated genes were previously implicated in clonal haematopoiesis of indeterminate potential (CHIP[21]; Supplementary Table 13). The associated variants in these CHIP genes were strongly correlated with age, and the proportion of reads supporting the variant in putative heterozygotes was often less than 35% or greater than 65% (Supplementary Table 14), consistent with these associations being driven by somatic mutations identified through exome sequencing of blood-derived DNA.

### Associations in non-European ancestries

We next investigated the extent to which associations identified in the European cohort were shared across other ancestries. To do so, we performed association analyses using WES data for 10,348 individuals of South Asian (SAS), 9,089 individuals of African (AFR) and 2,217 individuals of East Asian (EAS) ancestry from the UKB cohort. When we focused on the 564 non-redundant associations (that is, the strongest association per gene, 484 with a quantitative trait and 80 with a binary trait; Supplementary Table 6), we found that a large fraction of

associations was shared across ancestries for quantitative traits but less so for binary traits, probably owing to low power. For quantitative traits, effect sizes were directionally concordant for 83% of associations in individuals of SAS, 73% of associations in individuals of AFR and 74% of associations in individuals of EAS ancestry, increasing to more than 92% when considering associations with $P \leq 0.05$ (Extended Data Fig. 5a). For binary traits, consistent effects were observed for 61% of associations in SAS, 61% in AFR and 64% in EAS (Extended Data Fig. 5b). A similar pattern was observed when considering the full set of 8,865 associations (Supplementary Fig. 7). We then asked whether any new associations were discovered in non-European ancestries (Supplementary Data 3), and found four genes that were not discovered in the European-only analysis (*G6PD*, *HBQ1*, *OR51V1* and *RGS11*), all explained by previous established associations (Supplementary Table 15).

## Replication of rare variant associations

We sought to replicate associations using exome sequencing data from the Geisinger DiscovEHR cohort[22] ($n = 133,370$ individuals of European ancestry). As above, to facilitate interpretation of results, we focused on the non-redundant set of 564 gene–trait associations discovered in individuals of European ancestry in the UKB cohort (Supplementary Table 6) and determined whether a matching trait could be identified in DiscovEHR. Of the 279 gene–trait associations for which we attempted replication, 193 (69%; 28 with a binary trait, 165 with a quantitative trait) were nominally significant ($P \leq 0.05$) and directionally consistent (Supplementary Table 6), versus around 7 expected by chance ($279 \times 0.05 \times 0.5$). When considering only a subset of 212 gene–trait associations with at least 80% power for replication, the replication rate was 81% (172 of 212). Supplementary Data 2 provides replication results for all associations available in DiscovEHR (4,083 of 8,865), of which 70% were nominally significant and directionally consistent.

## Effect of burden test composition

As noted above, association of a phenotype with the burden of rare coding variants in a gene is a compelling way for human genetics to connect genes and disease[8]. As we show in the Supplementary Note, when we dissected burden associations in greater detail, we found that: (i) most (77% of 7,449) associations could not be detected in single-variant analyses (Supplementary Data 2), demonstrating that they were generally supported by multiple variants; (ii) burden tests that aggregated variants with a MAF of up to 1% identified a larger number of significant associations overall (Supplementary Table 16), but most of these remained significant after excluding variants with a MAF between 0.1% and 1% (Extended Data Fig. 6a), indicating that the greater yield is likely to be explained by the ability to capture in a single test association signals across a wide range of allele frequencies; and (iii) combining pLOFs and deleterious missense variants in the same test became progressively more valuable at more permissive MAF thresholds (Extended Data Fig. 6b). These results demonstrate the utility of performing a variety of burden tests for discovery of genetic associations.

## Enrichment of associations in GWAS loci

A major challenge for genetic association studies of complex traits is the identification of effector genes for the thousands of loci identified through GWAS[6]. To address the possibility that rare variant associations might help pinpoint effector genes, we performed a GWAS for each of the 492 traits with a rare variant association (Methods, Supplementary Data 1, Supplementary Fig. 4), and identified a total of 107,276 independent associations with common variants (hereafter 'GWAS sentinel variants'). As described in greater detail in the Supplementary Note, by combining results from the GWAS and the WES data, we found that:

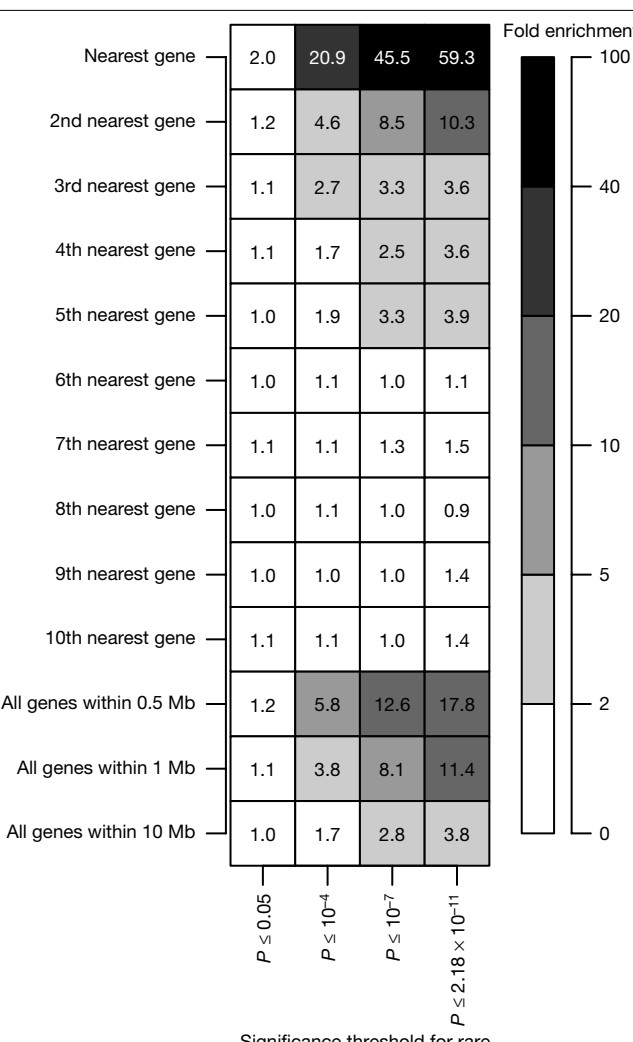

**Fig. 1 | Enrichment of rare variant associations among genes located in GWAS loci.** We tested whether genes located in GWAS loci were more likely to have significant associations with a burden of rare variants when compared to genes elsewhere in the genome. We considered four different significance thresholds to define significant burden associations ($P \leq 0.05$, $P \leq 10^{-4}$, $P \leq 10^{-7}$ and $P \leq 2.18 \times 10^{-11}$) and considered 13 different gene-sets, from all genes located within 10 Mb of, to only the nearest gene to, the GWAS sentinel variants. The enrichment of significant associations was greatest when considering the nearest gene to GWAS sentinel variants, reaching 59.3-fold (95% CI 51.8 to 68.2, $P < 10^{-300}$) when considering a significance threshold of $P \leq 2.18 \times 10^{-11}$. Results are based on the analysis of a pruned set of 188 traits (101 binary traits and 87 quantitative traits; see Methods for details).

(i) rare variant associations were often within 1 Mb of a GWAS sentinel variant for the same trait (6,564 of 8,865, 74%; Extended Data Fig. 4a); (ii) most rare variant associations (8,059 of 8,865, 91%) remained significant at $P \leq 2.18 \times 10^{-11}$ when we conditioned on GWAS common variant signals (Extended Data Fig. 4c, Supplementary Table 17, Supplementary Data 2); (iii) significant rare variant associations (after conditioning on GWAS signals) were 11.4-fold (95% CI 10.1 to 13.0, $P < 10^{-300}$) more common in genes located within 1 Mb of a GWAS peak, with enrichment reaching 59.4-fold (95% CI 51.8 to 68.2) when we focused only on genes nearest to GWAS sentinel variants (Fig. 1). These results show strong overlap between common variant signals from GWAS and rare variant signals from exome-wide association studies, suggesting that rare variant burden signals will identify effector genes for thousands of GWAS loci.

## Effector genes of GWAS signals

To illustrate the relevance of the findings described above, we highlight 168 genes for which a significant rare variant association ($P \leq 2.18 \times 10^{-11}$ after conditioning on common variants) was observed in the gene nearest to the GWAS sentinel variant (Supplementary Table 18), indicating that these are very likely to be effector genes that underlie the GWAS signal. As an example, we found 82 GWAS signals for serum levels of vitamin D (Extended Data Fig. 7a), and for 5 of these the burden of rare protein-altering variants in the gene nearest the GWAS peak (*DHCR7*, *FLG*, *GC*, *ANGPTL3* and *HAL*) was also associated with vitamin D levels (Extended Data Fig. 7b). Of these, we highlight the association with *HAL*, which has not to our knowledge been previously reported. The first step of vitamin D synthesis occurs in the skin and requires ultraviolet (UV) light. *HAL* is likely to have a role in this step because it encodes an enzyme that converts histidine into trans-urocanic acid, a major UV-absorbing chromophore that accumulates in the stratum corneum[23]. Inactivation of HAL is therefore expected to decrease the ability of the outermost layer of the epidermis to block UV light. Consistent with this possibility, we found that a burden of rare pLOF and deleterious missense variants in *HAL* was associated with higher levels of vitamin D, greater ease of skin tanning and higher risks of actinic keratosis and non-melanoma skin cancer (Supplementary Table 19). These findings were supported by trait-lowering associations with a common variant (rs10859995:C, 58% frequency) that co-localizes (LD $r^2 = 0.97$) with an expression quantitative trait locus (rs3819817:T) that increases the expression of *HAL* in skin tissue[14] (Extended Data Fig. 7c). These results implicate *HAL* in both vitamin D levels and skin cancer and highlight an allelic series that includes rare loss-of-function protein-altering variants (trait-increasing) as well as common expression-increasing non-coding variants (trait-lowering).

## Associations with brain imaging traits

The brain imaging component of UKB at present includes 2,077 phenotypes derived from magnetic resonance imaging (MRI) for 36,968 individuals. We analysed these data separately given the large number of traits and the relatively smaller sample size, testing the association with rare variants conditional upon GWAS signals as described above. We found 84 associations at $P \leq 2.18 \times 10^{-11}$ with 6 genes (Supplementary Table 20): *AMPD3*, *GBE1*, *PLD1*, *PLEKHG3*, *STAB1* and *TF*. Of these, we highlight the association between lower grey–white matter contrast (GWC) measures across a diffuse set of brain regions and a deleterious missense variant in *PLD1* (rs149535568, Gly237Cys, 196 carriers; effect = −0.49 s.d. units, 95% CI −0.62 to −0.35, $P = 1.4 \times 10^{-12}$), an enzyme that catalyses the hydrolysis of phosphatidylcholine to phosphatidic acid and choline, which has been shown to have a role in synaptogenesis[24]. GWC is a measure of blurring between the boundaries of grey- and white-matter brain compartments and is thought to be an indicator of local variations in tissue integrity and myelin degradation, increasing water content in the white matter, or iron deposition[25]. Lower GWC is associated with ageing and lower indices of cognition[26], as well as an increased rate of conversion from mild cognitive impairment to dementia[27]. Related to this finding, among an additional 46 genes with sub-threshold associations with brain imaging phenotypes ($P \leq 10^{-7}$; Supplementary Table 21), 4 genes had large trait-lowering effects on GWC, including 2 that have clear roles in the formation and maintenance of myelin—*GJC2*[28] and *UGT8*[29]—consistent with the association between variants that disrupt the function of these genes and lower GWC. In contrast, the strongest trait-increasing and putatively protective association with GWC was with a deleterious missense variant in *ST6GALNAC5* (rs756654226, Val135Ala, 9 carriers; effect = 1.7 s.d. units, 95% CI 1.1 to 2.4, $P = 8.2 \times 10^{-8}$), a gene that catalyses the biosynthesis of ganglioside from GM1b in the brain[30]. This aligns with current evidence that the relative abundance of specific gangliosides in the brain changes with age and in common neurological conditions[31]. We discuss notable associations with other genes (*GBE1*, *PLEKHG3*, *STAB1* and *TF*) in the Supplementary Note.

## Beyond 500,000 exomes

In our evaluation of the first 49,960 exomes sequenced from UKB participants[3], we used a beta-binomial model to predict the number of genes that would contain heterozygous pLOF variants when considering exome data for all 500,000 study participants. At current sample sizes, the observed and predicted numbers match closely (for example, 15,289 observed versus 15,613 predicted genes with at least 50 heterozygous pLOF carriers; Supplementary Table 22). Using our current dataset as a baseline (including all ancestries), we extended our projections to estimate the number of genes containing rare pLOFs (MAF ≤ 1%) when exome sequence data become available for 5 million individuals: we predict that 18,035, 17,853 and 8,376 genes will have at least 50, 100 and 500 heterozygous pLOF carriers, respectively (Supplementary Table 22, Extended Data Fig. 8a). Similarly, we predict that 2,630, 997 and 529 genes will have at least 10, 50 and 100 homozygous pLOF carriers, respectively, when considering 5 million sequenced individuals.

The UKB cohort consists primarily of individuals of European ancestry, and so an important question is whether these projections also apply to populations that are more ancestrally diverse. To address this, we predicted the number of pLOF carriers expected in 5 million individuals on the basis of (i) 46,000 individuals of European ancestry from the UKB; and (ii) 46,000 individuals from the UKB, including 23,000 of European ancestry and 23,000 individuals of other ancestries (10,000 of South Asian, 9,000 of African, 2,000 of East Asian, 1,000 of Hispanic or Latin American and 1,000 of admixed ancestry). We found that projections based on the more diverse set of samples were slightly higher than the estimates from the European-only dataset (Extended Data Fig. 8b).

## Whole-genome sequencing and imputation

In the coming years, we expect whole-genome sequence data to be available for all UKB participants, enabling analyses of rare variation to be extended to the remainder of the genome. Our data enable an early assessment of the value of that upcoming resource for genotype imputation, a widely used strategy for increasing the power, completeness and interpretability of array-based association studies[32]. We phased exome variants onto genotyping array haplotypes for 400,000 individuals, and then used this reference panel to impute exome variants into an array-only target dataset of 50,000 individuals. When reference and target datasets were well matched in ancestry, imputation accuracy remained high ($r^2 \geq 0.5$) for alleles present in at least 5 reference-panel haplotypes, enabling imputation down to an allele frequency of around 0.025%, 0.005% and 0.0005% in panels with around 10,000, 50,000 or 400,000 sequenced individuals (Supplementary Table 23, Fig. 2). As expected, imputation accuracy was lower, but still very good, when reference panel and target samples were less well matched in ancestry (Supplementary Table 23). Using reference panels of different sizes, we observed rapid increases in the ability to impute rare variants with larger panels and thus expect that even rarer variants will be imputable as reference panel sizes grow to 400,000 individuals and beyond (Extended Data Fig. 9, Supplementary Fig. 9).

## Discussion

We report the completion of exome sequencing for 454,787 UKB participants. Our dataset now includes an average of more than 600 coding variants per gene (including around 50 pLOFs per gene on average). In addition to enabling studies of mutation patterns and human demography[33], our dataset represents a major advance towards the

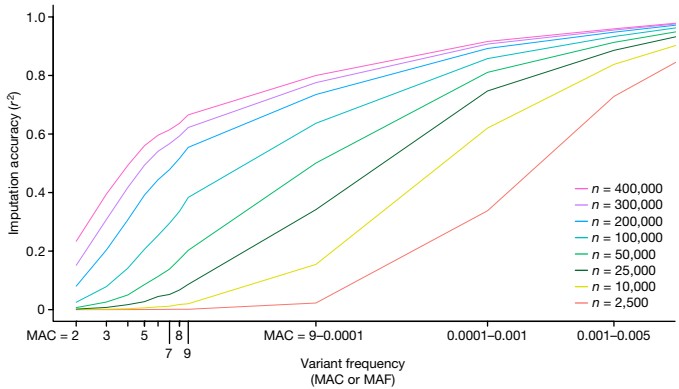

**Fig. 2 | Imputation of rare variants from exome sequencing.** Imputation accuracy ($r^2$, $y$ axis) is shown as a function of the variant allele frequency ($x$ axis; minor allele count (MAC) for ultra-rare variants; MAF for variants with MAF > $10^{-4}$) and the number of individuals ($n$) included in the reference panel (different lines). Full results are provided in Supplementary Table 23.

goal of understanding the health consequences of modifying each gene in the genome. In our preliminary analyses, we identify associations with health outcomes for pLOF and likely deleterious variation in 564 genes. These findings suggest new biological functions for many genes and potential therapeutic strategies, whether through enzyme replacement, therapeutic blockade or other modalities. All the data we generated are being made available to the UKB scientific community, and the combined creativity and efforts of this community will surely expand on these initial analyses.

The following caveats (expanded in the Supplementary Discussion) should be considered when interpreting our results. First, a small number of potentially low-quality variants may be included in the analysis, but our stringent significant thresholds and demonstrated replicability of most results suggest that this is not a widespread phenomenon. Second, disentangling mechanisms in genes associated with multiple traits will require careful follow-up analyses to distinguish situations in which a gene affects multiple traits directly from those in which additional signals are shadows of association with one trait. Third, while we focused on burden tests that could identify genes for which all pLOF or deleterious missense variants have a similar effect direction, additional association signals may be identified in genes that contain both trait-increasing and trait-lowering rare variants using alternative approaches such as SKAT[34]. In addition to these limitations, there are additional challenges that must be addressed with new samples and data: (i) there is limited genetic diversity among UKB participants and we expect that additional insights will become possible as more diverse samples are sequenced, particularly including insights that are relevant to the genetic disease burden specific to non-European individuals; (ii) although self-report questionnaires and electronic health records provide a very scalable way to phenotype hundreds of thousands of individuals, they naturally entail some misclassification—particularly when compared to more laborious and targeted phenotyping protocols; and (iii) given the very limited availability of complete nuclear families, it is not practical to carry out focused analyses of de novo variation, which has been shown to be especially important for several neurodevelopmental traits.

Accomplishing our original goal of understanding the health consequences of genetic variation in each human gene is likely to require sequencing millions of well-characterized and diverse individuals. In our view, our results not only show that this goal is within reach, but also suggest that sequencing 5 million individuals would enable the identification of more than 500 heterozygous LOF carriers for around 15,000 genes—that is, for the great majority of human protein-coding genes. It is our hope that these results and dataset will help provide the

impetus and urgency for generating these new datasets that combine health and variation data on millions of individuals.

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

# Methods

## Ethical approval and informed consent
Ethical approval for the UK Biobank was previously obtained from the North West Centre for Research Ethics Committee (11/NW/0382). The work described herein was approved by the UK Biobank under application no. 26041. Ethical approval for DiscovEHR analyses was provided by the Geisinger Health System Institutional Review Board under project no. 2006-0258. Informed consent was obtained for all study participants.

## Exome sequencing
**Sample preparation and sequencing.** We have previously described in detail the approach used at the Regeneron Genetics Center to perform exome sequencing in DNA samples from the UK Biobank study[3]. In brief, genomic DNA samples were transferred to the Regeneron Genetics Center from the UK Biobank and stored in an automated sample biobank at −80 °C before sample preparation. DNA libraries were then created by enzymatically shearing DNA to a mean fragment size of 200 base pairs, and a common Y-shaped adapter was ligated to all DNA libraries. Unique, asymmetric 10-base-pair barcodes were added to the DNA fragment during library amplification to facilitate multiplexed exome capture and sequencing. Equal amounts of sample were pooled before overnight exome capture, with a slightly modified version of IDT's xGen probe library. The initial 50,000 samples were processed with IDT 'lot 1' and all other samples with 'lot 2'. The captured DNA was PCR-amplified and quantified by quantitative PCR. The multiplexed samples were pooled and then sequenced using 75-base-pair paired-end reads with two 10-base-pair index reads on the Illumina NovaSeq 6000 platform using S2 (first 50,000 samples) or S4 (all other samples) flow cells. We sequenced all samples delivered to us by the UK Biobank. A portion of samples (about 30,000) could not be delivered because of the COVID-19 pandemic.

**Variant calling and quality control.** Sample read mapping and variant calling, aggregation and quality control were performed using the SPB protocol described previously[3]. In brief, for each sample, NovaSeq WES reads are mapped with BWA MEM to the hg38 reference genome. Small variants are identified with WeCall and reported as per-sample gVCFs. These gVCFs are aggregated with GLnexus into a joint-genotyped, multi-sample project-level VCF (pVCF). SNV genotypes with read depth (DP) less than 7 and indel genotypes with read depth less than 10 are changed to no-call genotypes. After the application of the DP genotype filter, a variant-level allele-balance filter is applied, retaining only variants that meet either of the following criteria: (i) at least one homozygous variant carrier; or (ii) at least one heterozygous variant carrier with an allele balance (AB) greater than the cut-off (AB ≥ 0.15 for SNVs and AB ≥ 0.20 for indels). Samples showing disagreement between genetically determined and reported sex ($n = 279$), high rates of heterozygosity or contamination (estimated with the VerifyBamId tool, specifically with a FREEMIX score > 5%) ($n = 287$), low sequence coverage (less than 80% of targeted bases achieving 20× coverage) ($n = 2$) or genetically identified sample duplicates ($n = 721$ total samples), and WES variants discordant with genotyping chip ($n = 449$), were excluded. A total of 633 samples failed quality control in multiple categories, resulting in 1,105 individuals being excluded. An additional 16 samples were removed for participants who withdrew from the study. The remaining 454,787 samples were then used to compile a pVCF for downstream analysis, using the GLnexus joint genotyping tool.

**Ancestry assignment.** We used array data released by the UK Biobank study to determine continental ancestry super-groups (African (AFR), Hispanic or Latin American (HLA, originally referred to as 'AMR' by the 1000 Genomes Project), East Asian (EAS), European (EUR) and South Asian (SAS)) by projecting each sample onto reference principal components calculated from the HapMap3 reference panel. In brief, we merged our samples with HapMap3 samples and kept only SNPs in common between the two datasets. We further excluded SNPs with MAF < 10%, genotype missingness > 5% or Hardy–Weinberg equilibrium test $P < 10^{-5}$. We calculated principal components (PCs) for the HapMap3 samples and projected each of our samples onto those PCs. To assign a continental ancestry group to each non-HapMap3 sample, we trained a kernel density estimator (KDE) using the HapMap3 PCs and used the KDEs to calculate the likelihood of a given sample belonging to each of the five continental ancestry groups. When the likelihood for a given ancestry group was greater than 0.3, the sample was assigned to that ancestry group. When two ancestry groups had a likelihood of greater than 0.3, we arbitrarily assigned AFR over EUR, HLA over EUR, HLA over EAS, SAS over EUR, and HLA over AFR. Samples were excluded from analysis if no ancestry likelihoods were greater than 0.3, or if more than three ancestry likelihoods were greater than 0.3 ($n = 1,205$).

**Generation of analysis-ready files.** The following steps were then taken to generate an analysis-ready Plink2 file set. First, we split exome data sample-wise into ancestral groups, defined as described above. Second, within ancestral groups, we excluded variants: (i) with missingness rate > 0.1; (ii) with Hardy–Weinberg equilibrium test $P < 10^{-15}$; or (iii) that were monomorphic. We also excluded samples with a missingness rate of greater than 0.1. After applying these filters, we generated ancestry-specific files in Plink2 PGEN format, which were then used for association analyses.

**Identification of low-quality variants from exome sequencing using machine learning.** In brief, we defined a set of positive control and negative control variants on the basis of: (i) concordance in genotype calls between array and exome sequencing data; (ii) Mendelian inconsistencies in the exome sequencing data; (iii) differences in allele frequencies between exome sequencing batches; (iv) variant loadings on 20 principal components derived from the analysis of variants with a MAF of less than 1%; (v) transmitted singletons. The model was then trained on up to 30 available WeCall/GLnexus site quality metrics, including, for example, allele balance and depth of coverage. We split the data into training (80%) and test (20%) sets. We performed a grid search with fivefold cross-validation on the training set to identify the hyperparameters that return the highest accuracy during cross-validation, which are then applied to the test set to confirm accuracy. This approach identified as low quality a total of 447,533 coding variants (3.7% of the 12 million total coding variants). These variants were flagged in (not removed from) downstream analyses.

## Variant annotation
Variants from WES were annotated as previously described[3]. In brief, variants were annotated using SnpEff, with the most severe consequence for each variant chosen across all protein-coding transcripts. Gene regions were defined using Ensembl release 85. Variants annotated as stop gained, start lost, splice donor, splice acceptor, stop lost or frameshift, for which the allele of interest is not the ancestral allele, are considered predicted LOF variants. Five annotation resources were used to assign deleteriousness to missense variants: SIFT[35]; PolyPhen2 HDIV and PolyPhen2 HVAR[36]; LRT[37]; and MutationTaster[38]. Missense variants were considered 'likely deleterious' if predicted deleterious by all five algorithms, 'possibly deleterious' if predicted deleterious by at least one algorithm and 'likely benign' if not predicted deleterious by any algorithm.

## Generation of gene burden masks
We aggregated rare variants for gene burden testing as previously described[39]. In brief, rare variants were collapsed by gene region, such that individuals who are homozygous reference for all variants

are considered homozygous reference, heterozygous carriers of any aggregated variant are considered heterozygous, and only minor allele homozygotes for an aggregated variant are considered as minor allele homozygotes. Genotypes were not phased to consider compound heterozygotes in burden testing. For each gene, we considered two categories of masks: a strict burden of rare pLOFs (M1) and a more permissive burden of rare pLOFs and likely deleterious missense variants (M3). For each of these groups, we considered five separate burden masks per gene, based on the frequency of the alternative allele of the variants that were screened in that group: MAF ≤ 1%, MAF ≤ 0.1%, MAF ≤ 0.01%, MAF ≤ 0.001%, and singletons only. Thus, overall, up to 10 burden tests were performed for each gene (although for some genes, the rarer burden tests may not have had enough (5) carriers across all samples, in which case the test was not performed). For the purposes of gene burden testing, the singleton mask includes minor allele homozygotes if no other variant carriers are observed in the dataset.

### Comparison with other large-scale resources

We compared variant statistics from UKB WES to two large, publicly available resources—gnomAD[5] v.3.1 and TOPMed[4] Freeze 8. For both studies, we restricted to 'PASS' variants only, and annotated each dataset as described for the UKB WES data. The comparison across datasets was restricted to synonymous, missense and pLOF variants only. We considered data from all ancestries.

### Imputation of unmeasured genotypes using the TOPMed reference panel

We used the following approach to generate imputed genotype data in the UKB study for variants discovered by the TOPMed consortium[4]. First, we began with the list of array variants previously used by UKB to perform HRC imputation. We removed all array variants that could not be successfully lifted over to GRCh38, leaving 655,665 variants. Second, we split the array data including 488,374 samples into 20 evenly sized, randomized batches for submission to the TOPMed imputation server. Third, we merged and concatenated the resulting VCF files from the imputation server into one dataset containing nearly 308 million imputed variants. We prepared this dataset for analysis by first splitting into batches of ancestry by continental super-groups, as previously described. We then filtered to variants that were predicted as functional, had a MAF value ≥0.0001 in the original TOPMed dataset, or passed the filters of MAF ≥ 0.0001 and INFO ≥ 0.1 within the dataset itself.

### Health- and behaviour-related phenotypes

Quantitative measures, clinical outcomes, survey and touch-screen responses, and imaging derived phenotypes were extracted from phenotypes available through the UK Biobank Data Showcase on 1 April 2020. To be considered a case for an ICD10-based phenotype, participants were required to have one or more of the following: (i) one or more diagnosis in inpatient Health Episode Statistics (HES) records; (ii) a cause-of-death diagnosis in death registry; (iii) two or more diagnoses in outpatient data (READ codes mapped to ICD10). Participants who did not meet the case definition for a given ICD10-based phenotype were either (i) excluded from the analysis if they had one diagnosis in the outpatient data; or (ii) included as controls if they had no diagnosis in the outpatient data. In total, data for 4,465 field IDs were downloaded from the UKB repository. We focused primarily on biomarkers, anthropometry and disease outcomes. As such, we excluded from analysis (i) most food and drink intake questions (except for coffee, tea and alcohol intake); (ii) quality control metrics (for example, volume or sample dilution information); (iii) geographic and environmental questions (for example, proximity to coast, pollution index); (iv) most measures pertaining to lifestyle or socioeconomic status (for example, number of cars owned, total household income); and (v) OPCS traits and any binary traits with fewer than 100 affected individuals. Furthermore, to reduce redundancy among binary traits, we excluded all 'No' responses from the analysis (for example, we analysed '22127_DD_asthma_1_Yes' but not '22127_DD_asthma_0_No'). In addition to HES and self-report data, we also generated custom phenotype definitions for a select number of diseases of interest, resulting in a total of 3,706 binary traits included in the study. For quantitative traits measured in multiple visits, we calculated the mean value across all visits for each participant and analysed only the resulting phenotype (for example, we analysed average height and not height measured at each visit). Only quantitative traits with data for more than 50,000 individuals, other than brain imaging phenotypes, were included in the analyses. We applied the following additional filters to systematically flag and exclude from analysis traits that were unlikely to be truly quantitative: (i) the mode for the trait was observed in 20% or more of samples (85 traits); (ii) the mode for the trait was observed in 0.5%–20% samples, but the number of unique values was relatively small (less than 100; 58 traits); or (iii) the mode for the trait was observed in 0.5%–20% samples, but the number of unique values was very large (more than 10,000; 9 traits), suggestive of a data error. The remaining 292 traits that passed quality control were normalized using a rank-based inverse-normal transformation.

### Brain imaging phenotypes

We analysed 2,158 phenotypes obtained by structural magnetic resonance imaging (MRI), diffusion MRI and task fMRI, downloaded from the UK Biobank Data Showcase on 1 April 2020. The traits were quantile-normalized and a matrix of confounds including age, sex, age-by-sex, head motion, head volume, head position, temporal imaging effects, imaging center and genetic PCs was regressed out of each trait before analysis, as described previously[40].

### Genetic association analyses

Association analyses were performed using the genome-wide regression test implemented in REGENIE[7], separately for data derived from exome-sequencing and TOPMed imputation. We included in step 1 of REGENIE (that is, prediction of individual trait values based on the genetic data) array variants with MAF > 1%, <10% missingness, Hardy–Weinberg equilibrium test $P > 10^{-15}$ and linkage disequilibrium (LD) pruning (1,000 variant windows, 100 variant sliding windows and $r^2 < 0.9$). We excluded from step 1 any SNPs with high inter-chromosomal LD, in the major histocompatibility (MHC) region, or in regions of low complexity. Of the 454,787 individuals with exome sequencing data, 413 did not have array data after quality control, and so these individuals were excluded from association analyses. For each trait, the leave-one-chromosome-out predictors obtained with step 1 were then included as covariates in step 2 for both the exome sequencing and TOPMed imputed data. The association model used in step 2 of REGENIE also included as covariates (i) age, age squared[2], sex, and age-by-sex; (ii) 10 ancestry-informative PCs derived from the analysis of a set of LD-pruned (50 variant windows, 5 variant sliding windows and $r^2 < 0.5$) common variants from the array data generated separately for each ancestry; and (iii) for the analysis of exome data, we additionally included an indicator for exome sequencing batch (6 IDT batches) and 20 PCs derived from the analysis of exome variants with a MAF between $2.6 \times 10^{-5}$ (roughly corresponding to a minor allele count (MAC) of 20) and 1% also generated separately for each ancestry. We corrected for PCs built from rare variants because previous studies demonstrated that PCs derived from common variants do not adequately correct for fine-scale population structure[41,42]. We tested associations with genes on chromosome X but not Y. For the non-pseudoautosomal regions of chromosome X, we used a dosage compensation model, with homozygous reference males coded 0, and hemizygous males coded 2.

Association analyses were performed separately for different continental ancestries defined based on the array data, as described above, analysing variants with an MAC of five or greater. Analysis of TOPMed

imputed data was only performed for 492 traits that had at least 1 significant rare variant association in the exome sequencing data.

## Estimating power to identify risk-lowering and risk-increasing associations

**Empirical power calculations.** We simulate genotype and phenotype data without population structure or relatedness, using the same sample size as that available for individuals of European ancestry ($n$ = 430,998). Markers are simulated independently with alleles drawn from Binomial(2, EAF) based on a given effect allele frequency (EAF) level. We use a logistic model to generate the binary trait:

$$\mathrm{logit}(p) = \mu + G\beta$$

in which $\mu$ is chosen to achieve a desired prevalence level $K$, $G$ is the genotype vector for the causal marker and $\beta = \log(\mathrm{OR})$ is the effect of the causal marker, and the trait is generated as $Y|p$ - Bernoulli($p$). We vary the EAF between 1%, 0.1%, 0.01% and 0.001%, and for each setting generate 10 marker replicates. To simulate a binary trait, we consider the disease prevalence $K$ at 10%, 1% or 0.1% and vary the OR between 1, 0.75, 0.5, 0.35, 0.2 and 0.01 for risk-lowering (protective) variants and 1, 1.5, 2, 5, 10, 20, 30, 40 and 50 for risk-increasing (predisposing) variants. For each simulation setting with 10 marker replicates, we generate 100 phenotypic replicates, which results in 1,000 replicates, and we perform association testing using REGENIE-FIRTH in which the $P$ value fallback threshold for Firth correction is set to 0.05. Empirical power was then estimated as the proportion of 1,000 simulation replicates with a $P$ value below a significance level $\alpha$ of $2.18 \times 10^{-11}$.

**Theoretical power.** For comparison, we computed theoretical power based on a logistic regression score test as previously described[43], where the non-centrality parameter $\eta$ is

$$\eta = \frac{2N_0 N_1 (p' - p)^2}{(N_0 + N_1)\overline{p}(1 - \overline{p})}$$

where $N_1$ and $N_0$ represent the number of cases and controls, respectively, $p$ is the EAF in controls (approximated by the EAF in the population), $p'$ is the EAF in cases and $\overline{p}$ is the EAF in the study (taken as a weighted average of the EAF in cases and controls).

## Leveraging associations with quantitative traits to identify protective associations with relevant diseases

We tested the association between rare variants and 292 quantitative traits, and then leveraged associations with these traits to identify protective associations with relevant diseases. The following four steps were taken to do this. First, for each quantitative trait, we determined whether higher or lower trait levels are associated with a beneficial effect on health. For example, higher bone mineral density is generally accepted to be associated with lower risks of osteoporosis and fractures and, similarly, lower eosinophil counts are associated with lower risks of asthma and atopic dermatitis. Of the 292 quantitative traits tested, for 85 there was consensus among a team of experts in diverse therapeutic areas on the directionality that is associated with beneficial health outcomes.

Second, among all rare variant associations with each of those 85 traits, we identified the subset for which the direction of effect on the trait was beneficial. For example, we identified rare variants that increased (not reduced) bone mineral density, and rare variants that reduced (not increased) eosinophil counts. We found 34 such traits with at least one directionally favourable rare variant association.

Third, we matched each of these 34 quantitative traits to a single relevant disease. We did this by estimating the genetic correlation between each trait and 357 disease outcomes (specifically, 3-digit ICD codes, expert-curated definitions, self-report and doctor-diagnosed diseases; we only considered diseases that had at least 1 rare variant association

at $P < 10^{-7}$), using LD score regression[44] and association results from the TOPMed-based GWAS described above. We used LD scores calculated for HapMap3 variants in individuals of European ancestry from the 1000 Genomes Project, with variant positions lifted over to genome build GRCh38. For each trait, we then identified any genetic correlations that were significant after correcting for the 357 tests performed ($P < 0.05/357 = 1.4 \times 10^{-4}$) and then, if any, selected the disease that had the most significant genetic correlation for follow-up analysis. In this way, we were able to match 33 of the 34 quantitative traits to a relevant disease.

Finally, for each gene with a significant ($P \le 2.18 \times 10^{-11}$) and directionally favourable effect on one of these 33 quantitative traits (for example, *IL33* pLOFs and association with lower eosinophil counts), we then determined if there was a consistent protective association with the matched disease (for example, *IL33* pLOFs and protection from asthma).

## Determining whether associations were likely to be attributable to somatic mutations

We found a small number of traits with (i) two or more genes with a rare variant association; and (ii) no GWAS common variant signals. For a subset of these traits, we noticed that the associated genes have been implicated in CHIP[21,45]. Therefore, we addressed the possibility that the observed associations with this small group of traits were explained by somatic mutations identified through exome sequencing of blood-derived DNA. To address this possibility, we (i) estimated the association between each variant (or burden test) and age, because the frequency of somatic (but not germline) mutations typically increases strongly with age; and (ii) counted the number of variant carriers for whom the proportion of sequencing reads supporting the presence of the alternative allele (that is, variant allele fraction) was less than 35% or more than 65%, which would be more consistent with the variant being of somatic than of germline origin.

## Replication in the DiscovEHR cohort

The Geisinger Health System (GHS) DiscovEHR cohort has been described previously[22]. In brief, DiscovEHR is a health-system-based cohort from central and eastern Pennsylvania (USA) with ongoing recruitment since 2006. A subset of 133,370 MyCode participants sequenced as part of the GHS–Regeneron Genetics Center DiscovEHR partnership and confirmed to be of European ancestry were included in this study. We attempted to replicate in DiscovEHR the most significant variant–trait association for each gene, as listed in Supplementary Table 6. We only considered associations for which the trait tested in the UKB cohort could be matched unambiguously to a trait available in the DiscovEHR cohort. To determine whether the DiscovEHR cohort provided adequate power to replicate an association discovered in the UKB, we carried out a winner's curse-corrected power analysis as described previously[40]. In brief, power to replicate a given trait-variant association in the DiscovEHR cohort at $P < 0.05$ was estimated based on the following parameters: (i) effect size in the UKB cohort (beta), after adjusting for winner's curse; (ii) standard error of the effect size in the DiscovEHR cohort; and (iii) sample size in DiscovEHR cohort. The same approach was used for quantitative and binary traits.

## Identification of rare variant associations that were independent of GWAS signals

For each of the 492 traits with at least one rare variant association at $P \le 2.18 \times 10^{-11}$, we (i) identified common variants independently associated with the trait at $P \le 10^{-7}$; and (ii) determined whether the rare variant associations remained significant after adjusting for the common variant signals.

To identify common variants independently associated with a given trait, we first performed a GWAS for that trait that included the same individuals used in the analysis of exome-sequencing data and common variants (MAF > 1%) imputed from TOPMed, as described above.

We then identified independent signals (in the autosomes and the X chromosome) using the approximate conditional analysis implemented in GCTA v.1.91.7[46]. To estimate linkage disequilibrium, we randomly sampled 10,000 individuals from the UK Biobank TOPMed imputed dataset, with dosages between 0 and 0.1 considered homozygote for the reference allele (genotype = 0), between 0.9 and 1.1 considered a heterozygote (genotype = 1), and between 1.9 and 2 considered a homozygote for the alternative allele (genotype = 2); all other dosages were assigned a missing genotype. We performed approximate conditional analysis using a window of 10 Mb, collinearity = 0.9 and variants with a MAF > 1%. We then retained all variants that had an association $P \leq 10^{-7}$ in the GCTA-cojo joint model. These independently associated variants were then included as covariates when analysing rare variants from exome sequencing data, as described below. We used $P \leq 10^{-7}$ to ensure that we included in the subsequent conditional analyses of exome sequencing data any common variant signals that were close to (but did not quite surpass) the more commonly used genome-wide significance threshold of $P \leq 5 \times 10^{-8}$. However, when reporting the number of independent common variant signals for each trait, we consider only the subset that had $P \leq 5 \times 10^{-8}$, to be consistent with previous studies. Overall, of the 492 traits for which we performed a GWAS, 429 had at least 1 common variant with $P \leq 10^{-7}$ and 421 had at least 1 common variant with a $P \leq 5 \times 10^{-8}$.

Having identified independent common variant signals for a given trait, we then tested whether rare variant associations remained significant after adjusting for those common variant signals. To this end, for each trait, we repeated the association analysis in REGENIE (step 2 only; we used the genome-wide predictors that were created in step 1 as part of the original analysis, which did not condition on any common variants) but now including as additional covariates the dosages for all common variants that were found to have an independent association with the trait, as described above. Associations that exceeded $P \leq 2.18 \times 10^{-11}$ in these conditional analyses were determined to be independent of the common variant signals. Conditional analyses were performed for 429 (out of 492) that had at least one GWAS signal at $P \leq 10^{-7}$. For the remaining 63 traits ( = 492–429), there were no common variants with $P \leq 10^{-7}$ and so for these traits rare variant signals were considered to be independent of GWAS signals.

## Number of rare variant associations expected to be found in GWAS loci by chance

We determined whether the number of rare variant associations that were found to be within 1 Mb of a GWAS signal (specifically 6,564 out of 8,865 associations) was greater than that expected by chance. The number expected by chance was estimated as $p \times k$, where $p$ is the proportion of significant associations among all association tests performed across the genome, considering all rare variants (individual variants and burden tests) and the 492 traits with at least one rare variant association; and $k$ is the number of association tests performed across variants located within 1 Mb of a GWAS signal, considering only the rare variant–trait pairs for the matching GWAS common variant–trait pair, as detailed below. Specifically, $p = a / n = 0.0000285$, given that $a = 8,865$; that is, the total number of rare variant associations with $P \leq 2.18 \times 10^{-11}$ across the 492 traits; and $n = 311,080,453$; that is, the total number of rare variant association tests performed across the 492 traits. In turn, $k$ was determined as follows: (i) for each of the 107,276 independent GWAS signals, we identified rare variants that were located within 1 Mb of the GWAS sentinel variant and that were tested for association with the same trait; (ii) for each trait, we then added the number of rare variants tested across all GWAS signals for that trait, removing duplicate entries, if any; and (iii) added the number of rare variant tests performed across all traits. Using this approach, we found that $k = 131,077,005$ tests. Therefore, the number of significant rare variant associations that were expected to be found within 1Mb of a GWAS signal by chance was $0.0000285 \times 131,077,005 = 3,736$.

## Determining the enrichment of rare variant associations among genes in GWAS loci

We used the following approach to determine whether genes located within 1 Mb of GWAS signals were more likely to have a significant rare variant association (specifically, a burden test with $P \leq 2.18 \times 10^{-11}$ after controlling for GWAS signals, to ensure that rare and common variant signals were independent) when compared to other genes in the genome. First, for each trait, we counted the number of genes that (i) were located within 1 Mb of a GWAS sentinel variant and had a significant rare variant association [a]; (ii) were located within 1 Mb of a GWAS sentinel variant and did not have a significant rare variant association [b]; (iii) were not located within 1 Mb of a GWAS sentinel variant and had a significant rare variant association [c]; and (iv) were not located within 1 Mb of a GWAS sentinel variant and did not have a significant rare variant association [d]. For a given trait, the fold-enrichment of significant rare variant associations among genes within 1 Mb of a GWAS signal was estimated as $(a/b)/(c/d)$. Second, to obtain an overall measure of enrichment across all traits, we used the Mantel–Haenszel approach to combine the trait-specific enrichment results (specifically, the 2-by-2 table defined by values $a$, $b$, $c$ and $d$), with significance of the overall estimate being determined by a chi-squared test. The GWAS signals considered in this analysis were located more than 10 Mb apart, to ensure that a given gene could only be matched to a single GWAS signal. We repeated this analysis for different gene sets (for example, genes located within 0.5 Mb of a GWAS signal; 10th nearest gene to a GWAS signal; nearest gene to a GWAS signal; and so on) and different thresholds to define significant rare variant associations ($P \leq 10^{-7}$, $P \leq 10^{-4}$ and $P \leq 0.05$). Of the 421 traits that had at least 1 gene with a significant rare variant association at $P \leq 2.18 \times 10^{-11}$ and also at least one GWAS signal at $P \leq 5 \times 10^{-8}$, we restricted this analysis to a subset of 188 traits (101 binary traits, 87 quantitative traits), obtained after excluding highly redundant traits (for example, there were 20 traits related to body mass, 14 traits related to bone mineral density, both absolute and relative blood cell counts, self-reported and ICD10-based diagnoses).

## Imputation of exome variants using a reference panel with array and exome variants

We used SNP array and exome sequencing data from the UK Biobank on 454,378 individuals. For SNP array data, we excluded variants that were not used during a previous round of phasing[2], resulting in 670,423 SNP array sites. For exome sequencing data, we excluded variants that had an MAC of one or that were flagged has potentially having low quality by the machine learning approach described above, resulting in 15,845,171 exome variants. We then phased these array and exome datasets as follows. First, we built a haplotype scaffold by phasing SNP array data with SHAPEIT4.2.0[47], phasing whole chromosomes at a time. We then phased the exome sequencing data onto the array scaffold in chunks of 10,000 variants, using 500 SNPs from the array data as a buffer at the beginning and end of each chunk. A consequence of this process is that when a variant appears in both the array and exome datasets, it is the data from the array dataset that are used.

The phased SNP array and exome sequencing dataset was split into 2 sets: a set of 404,378 reference panel individuals and a target set of 50,000 individuals. To systematically study the effect of reference panel size on imputation accuracy, we generated reference panels by using 2,500, 10,000, 25,000, 50,000, 100,000, 200,000, 300,000 and 400,000 individuals from the set of 404,378 individuals. Each reference panel was then used to impute exome variants using the SNP array data from the 50,000-sample target dataset. The imputation was carried out on chromosome 2 only in chunks of 20 Mb using IMPUTE5[48], which exhibits sub-linear scaling as reference panel size grows. We examined the sensitivity of these results to ancestry in two ways. First, by measuring imputation accuracy in ancestry-specific subsets of the 50,000 target dataset for the 400,00 reference panel

results (for example, only among individuals of South Asian ancestry). Second, we created a single reference panel of 300,000 individuals with principal component analysis (PCA)-derived European ancestry and who self-reported as 'White British', and a separate test dataset of 49,926 individuals with PCA-derived European ancestry who did not self-identify as 'White British'. This testing scenario is denoted 300,000 WB in Supplementary Table 23.

We measured imputation accuracy by comparing the imputed dosage genotypes to the true (masked) genotypes at exome variants. Markers were binned according to the MAF of the marker in either the reference panel or the full dataset of 454,374 individuals. In each bin, we report the squared correlation ($r^2$) between the concatenated vector of all the true (masked) genotypes at markers and the vector of all imputed dosages at the same markers. At the ultra-rare end of the frequency spectrum, we use individual values of MAC for the bins, instead of MAF.

We used imputation accuracy results obtained across different sizes of the reference panel (shown in Supplementary Table 23) to extrapolate performance at larger reference panel sizes. For each MAC or MAF bin we fit logistic curve models to the $r^2$ values at reference panel sizes $n = 50,000, 100,000, 200,000, 300,000$ and $400,000$ of the form $r^2 \approx c/(1 + \exp(-(a + b \times \log(n))))$. We tried two versions of this model: a two-parameter model with the asymptote ($c$) fixed at 1, and a three-parameter model that has the restriction that $c \leq 1$. Allowing the logistic curve to be parameterized on $\log(n)$ scale was important. We then used these curves to extrapolate to larger reference panel sizes up to $n = 1,000,000$. The resulting fitted curves from the two- and three-parameter models are shown in Extended Data Fig. 9a and Supplementary Fig. 9a, respectively, with associated 95% CI estimated using the delta method. To assess the accuracy of this approach, we repeated the process by excluding the $r^2$ value for $n = 400,000$, and then used the logistic curve to predict $r^2$ at $n = 400,000$ (shown by the blue dot on each plot in Extended Data Fig. 9a, Supplementary Fig. 9a). We then aggregated the results into single plots (Extended Data Fig. 9b, Supplementary Fig. 9b) that show both the results of our imputation experiments together with the extrapolated values. The two-parameter logistic model seems to overestimate imputation accuracy in some MAC/MAF bins. This is especially evident when looking at the $n = 400,000$ prediction (Extended Data Fig. 9b). The three-parameter logistic model seems to perform better for the $n = 400,000$ prediction except for the MAC = 2 bin, in which the predictions seem too high and inconsistent with predictions at higher bins.

### Prediction of pLOF carriers beyond 500,000 exomes
We estimated the number of pLOF carriers expected to be observed in one and five million sequenced samples using a mixture model of beta-binomial distributions, as previously described[3]. Model parameters were estimated using heterozygous and homozygous pLOF counts per autosomal gene in 454,787 exomes spanning all ancestries.

### Reporting summary
Further information on research design is available in the Nature Research Reporting Summary linked to this paper.

### Data availability
Individual-level sequence data have been deposited with the UK Biobank and are freely available to approved researchers, as has been done with other genetic datasets to date. Individual-level phenotype data are also available to approved researchers for the surveys and health-record datasets from which all of our traits are derived. Instructions for access to UK Biobank data are available at https://www.ukbiobank.ac.uk/enable-your-research. Full details for the significant trait associations with rare variants described in this study are provided in Supplementary Data 2, 3. Summary statistics for the rare variants tested in this study are also available in the GWAS Catalog

(accession IDs are in Supplementary Data 4 and are listed separately for single variants and burden tests). The HapMap3 reference panel was downloaded from ftp://ftp.ncbi.nlm.nih.gov/hapmap/. GnomAD v3.1 VCFs were obtained from https://gnomad.broadinstitute.org/downloads. VCFs for TOPMED Freeze 8 were obtained from https://bravo.sph.umich.edu/freeze8/hg38/downloads. LD scores from 1000 Genomes Project were downloaded from https://data.broadinstitute.org/alkesgroup/LDSCORE/.

### Code availability
The association analysis package used to perform all genetic associations is publicly available at https://github.com/rgcgithub/regenie.

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

**Acknowledgements** This research has been conducted using the UK Biobank Resource (project 26041). The authors thank everyone who made this work possible, particularly the UK Biobank team, their funders, the professionals from the member institutions who contributed to and supported this work, and most especially the UK Biobank participants, without whom this research would not be possible. The exome sequencing was funded by the UK Biobank Exome Sequencing Consortium (Bristol Myers Squibb, Regeneron, Biogen, Takeda, Abbvie, Alnylam, AstraZeneca and Pfizer). Ethical approval for the UK Biobank was previously obtained from the North West Centre for Research Ethics Committee (11/NW/0382). The work described herein was approved by the UK Biobank under application number 26041. Approval for DiscovEHR analyses was provided by the Geisinger Health System Institutional Review Board under project number 2006-0258. Informed consent was obtained for all study participants.

**Author contributions** Conceptualization: J.D.B., A.H.L., H.M.K., G.C., L.A.L., J. Marchini, G.R.A., A.B. and M.A.R.F. Data curation: A.Y., N.B., V.R., E.S. and M.N.C. Data generation: X.B., A.H., W.J.S., J.G.R. and J.D.O. Formal analysis: J.D.B., A.H.L., A.M., D.S., J. Mbatchou, A.E.L., C.E.G., J.A.K., D.L., M.D.K., A.D., L.G., C.B. and M.A.R.F. Funding acquisition: A.R.S., G.Y., J.D.O., J.G.R., G.R.A. and A.B. Methodology: J. Marchini, H.M.K., G.R.A. and M.A.R.F. Project administration: J. Mighty, M.J. and L.M. Resources: S.B., S.L., E.M., L.H. and J.G.R. Software: J. Mbatchou and J. Marchini. Supervision: E.J., J.D.O., M.N.C., J.G.R., H.M.K., J. Marchini, A.B., G.R.A. and M.A.R.F. Visualization: J.D.B., A.H.L., K.W., M.D.K., C.E.G. and M.A.R.F. Writing (original draft): J.D.B., A.H.L., J. Marchini, G.R.A. and M.A.R.F. Writing (review and editing): E.J., H.M.K., G.Y., A.R.S. and A.B.

**Competing interests** J.D.B., A.H.L., A.M., D.S., J. Mbatchou, C.E.G., D.L., A.E.L., S.B., A.Y., N.B., M.D.K., A.D., S.L., C.B., X.B., A.H., E.M., L.G., K.W., J.A.K., V.R., J. Mighty, M.J., L.M., G.C., E.J., L.H., W.J.S., A.R.S., L.A.L., J.D.O., M.N.C., J.G.R., G.Y., H.M.K., J. Marchini, A.B., G.R.A. and M.A.R.F. are current employees and/or stockholders of Regeneron Genetics Center or Regeneron Pharmaceuticals.

**Additional information**
**Correspondence and requests for materials** should be addressed to Gonçalo R. Abecasis or Manuel A. R. Ferreira.

**a   80 genes for which the lead association was with a binary trait**

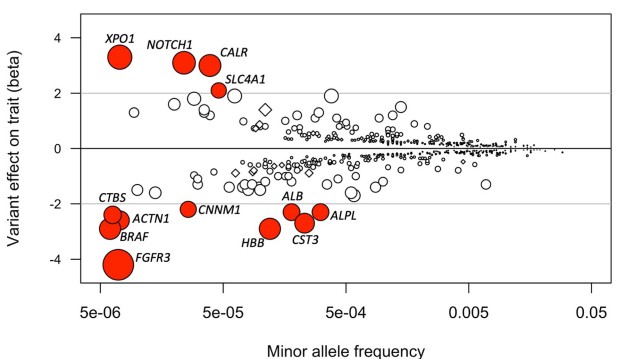

**Genes with an odds ratio >100:**

| Gene | Most associated binary trait |
|---|---|
| CHD2 | Chronic lymphocytic leukemia of B-cell type |
| COL1A1 | Bone disorder |
| ENG | Hereditary hemorrhagic telangiectasia |
| MEN1 | Hyperparathyroidism |
| NF1 | Benign neoplasm of peripheral nerves |
| NFKBIE | Chronic lymphocytic leukemia of B-cell type |
| PKD2 | Cystic kidney disease |
| SERPINC1 | Coagulation defects |
| UMOD | Chronic kidney disease |

**b   484 genes for which the lead association was with a quantitative trait**

**Genes with |effect| >2:**

| Gene | Most associated quantitative trait |
|---|---|
| ACTN1 | Platelet count |
| ALB | Albumin |
| ALPL | Alkaline phosphatase |
| BRAF | Neutrophil count |
| CALR | Platelet count |
| CNNM1 | Aspartate aminotransferase |
| CST3 | Cystatin C |
| CTBS | Peak expiratory flow |
| FGFR3 | Height |
| HBB | Mean corpuscular volume |
| NOTCH1 | Lymphocyte count |
| SLC4A1 | Reticulocyte percentage |
| XPO1 | Lymphocyte count |

**Extended Data Fig. 1 | Lead trait associations for 564 genes with a rare variant association at $P \leq 2.18 \times 10^{-11}$. a**, Associations with binary traits. **b**, Associations with quantitative traits. In red (and table): associations with odds ratio >100 for binary traits and |effect| > 2 for quantitative traits. Diamonds show associations that were no longer significant after accounting for nearby GWAS signals.

**a** Protective associations (odds ratio <1)

**b** Predisposing associations (odds ratio >1)

**Extended Data Fig. 2 | Power to identify associations with rare variants in the analysis of 430,998 participants of European ancestry from the UK Biobank. a**, Protective associations (i.e. with an odds ratio <1). **b**, Predisposing associations (i.e. with an odds ratio >1). Power was estimated using asymptotic theory (broken lines) and also through simulations (solid lines), separately for variants with an effect allele frequency (EAF) of 1% (purple), 0.1% (blue), 0.01% (green) and 0.001% (yellow). Power to identify protective associations was low because identification of rare variants that reduce disease risk typically requires very large numbers of cases, and population cohorts like that ascertained by the UK Biobank study typically include many more unaffected than affected individuals for each disease.

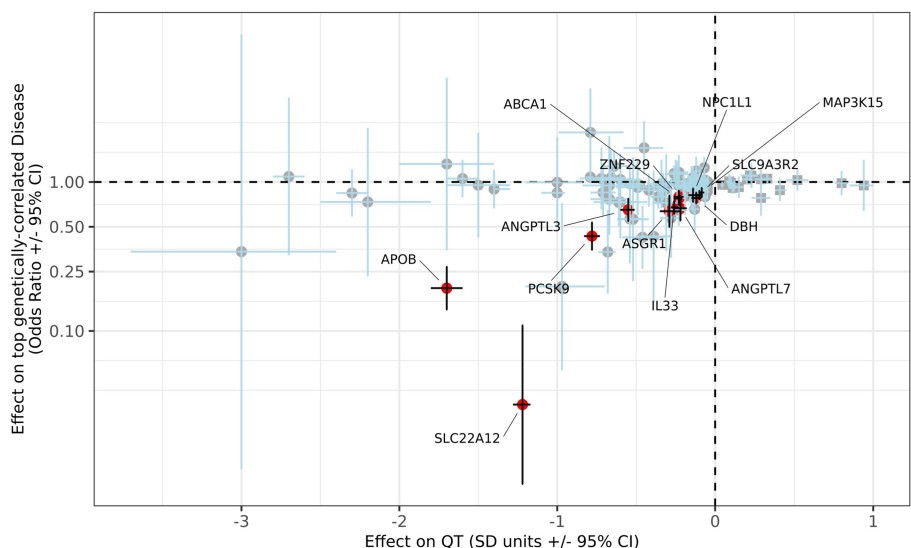

**a**

**b**

| Gene | Quantitative trait | Disease (genetic correlation, $r_g$) |
|---|---|---|
| **KNOWN** | | |
| *DBH* | ↓ DBP | ↓ Hypertension (0.78) |
| *ABCA1* | ↓ Cholesterol | ↓ High Cholesterol (0.52) |
| *ANGPTL3* | ↓ Triglycerides | ↓ Hyperlipidemia (0.54) |
| *ANGPTL7* | ↓ IOP - Goldman | ↓ Glaucoma (0.66) |
| *APOB* | ↓ Apolipoprotein B | ↓ High Cholesterol (0.66) |
| *ASGR1* | ↓ Apolipoprotein B | ↓ High Cholesterol (0.66) |
| *IL33* | ↓ Eosinophil % | ↓ Asthma (0.37) |
| *NPC1L1* | ↓ LDL | ↓ High Cholesterol (0.60) |
| *PCSK9* | ↓ Apolipoprotein B | ↓ High Cholesterol (0.66) |
| *SLC22A12* | ↓ Urate | ↓ Gout (0.88) |
| *ZNF229* | ↓ Apolipoprotein B | ↓ High Cholesterol (0.66) |
| **NOVEL** | | |
| *SLC9A3R2* | ↓ SBP | ↓ Hypertension (0.82) |
| *MAP3K15* | ↓ HbA1c | ↓ T2D (0.63) |

**Extended Data Fig. 3 | Genes for which a rare variant had a favourable effect on a quantitative trait ($P \leq 2.18 \times 10^{-11}$) and also a protective association with a genetically correlated disease. a**, The x- and y-axes show the effect of the rare variant (listed in Supplementary Table 12) on the quantitative trait and genetically correlated disease, respectively. **b**, Thirteen genes for which the disease association was significant after correcting for multiple testing ($P \leq 0.05/129 = 3.8 \times 10^{-4}$; also shown in red in panel **a**).

**a**

| Analysis | N traits | | N significant associations | |
|---|---|---|---|---|
| | Tested | With association signal | With rare variants (N, % near GWAS signal) | With common variants (N, % near ExWAS signal) |
| ExWAS | 3,994 | 492 | 8,865 (6,564, 74%) | - |
| GWAS | 492 | 421 | - | 107,276 (7,546, 7%) |

**b**

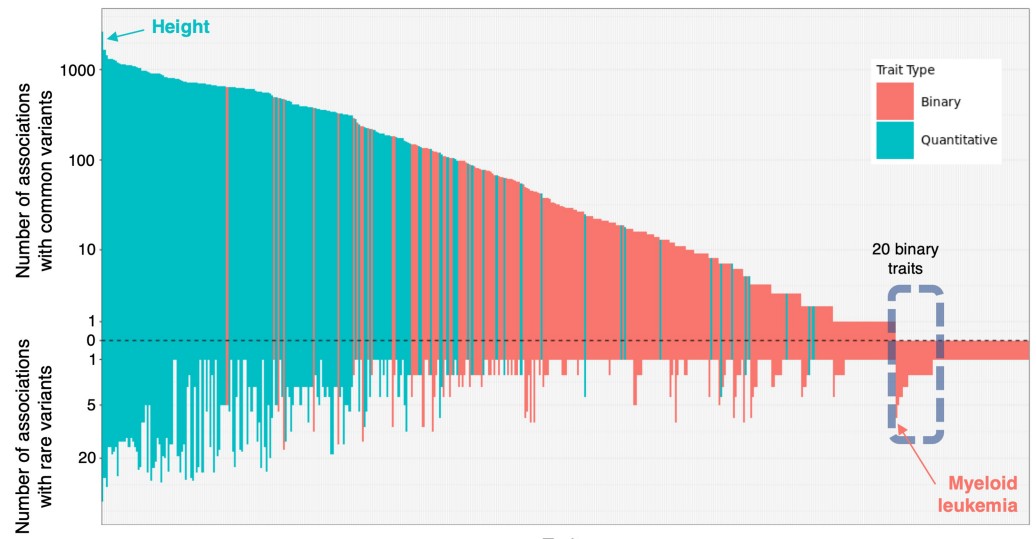

**c**

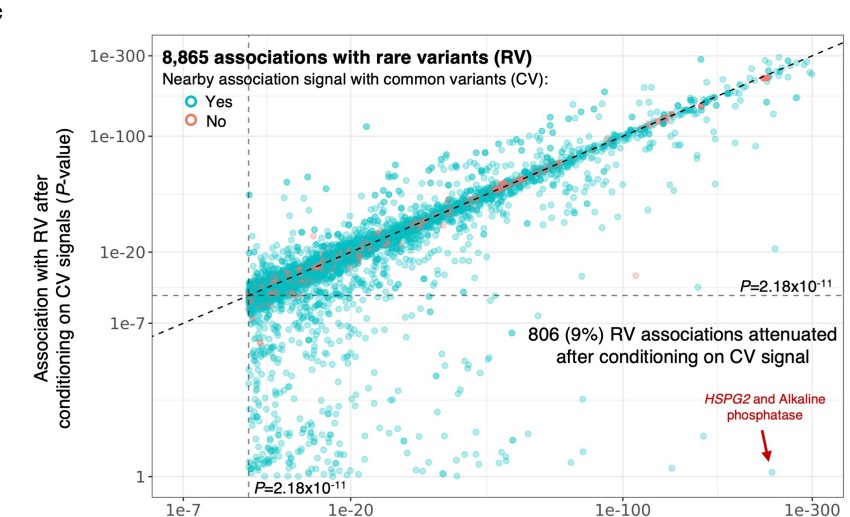

**Extended Data Fig. 4** | See next page for caption.

**Extended Data Fig. 4 | Associations with common and rare variants in individuals of European ancestry. a**, Number of traits tested and genetic associations discovered in UKB 450K. An exome-wide association study (ExWAS) was performed for 3,994 traits, of which 492 had at least 1 gene with a rare variant (RV) association at $P \leq 2.18 \times 10^{-11}$. Across all 492 traits, we identified 8,865 significant RV associations, a list that includes redundant associations arising from having tested multiple (often correlated) variants and traits per gene. The 8,865 associations (including 6,564 or 74% located within 1 Mb of a GWAS signal) reduced down to (i) 2,283 associations when selecting only the most significant association per gene per trait (Supplementary Data 2); and (ii) 564 associations when selecting only the most significant association per gene (Supplementary Table 6). For each of the 492 traits with at least 1 RV association, we performed a genome-wide association study (GWAS) using TOPMed data for the same individuals included in the ExWAS. Of the 492 traits, 421 had at least 1 common variant (CV) signal at $P < 5 \times 10^{-8}$. Independent CV associations were identified for each trait using approximate conditional analysis, and then the number of independent associations was summed across all traits, for a total of 107,276 associations (including 7,546 or 7% that were located within 1 Mb of an ExWAS signal). **b**, Top half of the figure shows number of independent CV signals (MAF>1% and conditionally independent) per trait, from the TOPMed GWAS. Bottom half of the figure shows number of genes with a RV association for the same trait from ExWAS. The x-axis shows all 492 traits that had 1 or more genes with a RV association, sorted by the number of CV signals, with ties in turn sorted by number of genes with a RV association. Traits that did not have RV signals (3,994-492 = 3,506) are not shown on this plot. Twenty binary traits that had 2 or more genes with a RV association but no CV signals from the GWAS are highlighted by the dashed box and listed in Supplementary Table 13. **c**, This panel shows associations with RV before (x-axis) and after (y-axis) accounting for the effect of CV signals. Of the 8,865 RV associations, 806 (9%) were no longer significant at $P < 2.18 \times 10^{-11}$ after accounting for CV signals (see also Supplementary Table 17). The highlighted association between *HSPG2* and alkaline phosphatase is an example of an association that was greatly attenuated after controlling for the effect of CV signals (regional association plots shown in Supplementary Fig. 8).

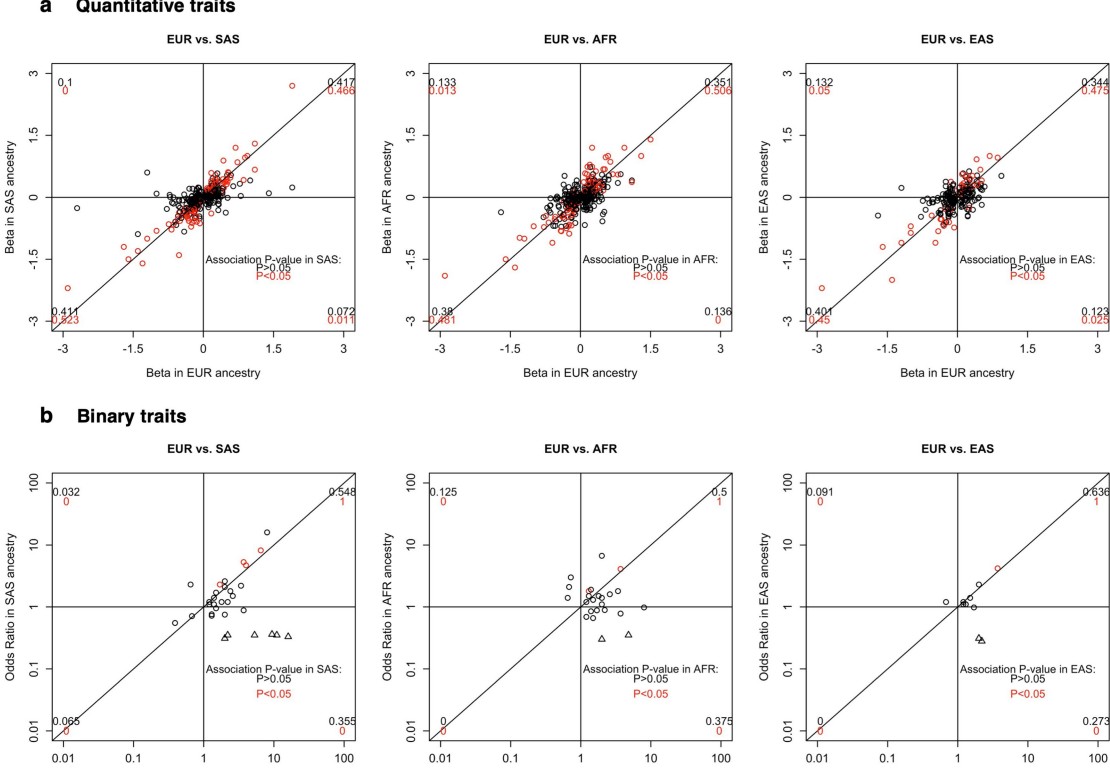

**Extended Data Fig. 5 | Comparison of effect sizes across ancestries for 564 lead associations identified in Europeans.** For each of the 564 genes with at least 1 rare variant association in individuals of European (EUR) ancestry, we selected the most significant association (484 with a quantitative trait, 80 with a binary trait; see Supplementary Table 6) and then compared the effect size estimated in Europeans with that estimated in individuals of South Asian (SAS), African (AFR) and East Asian (EAS) ancestry, if available. **a**, Of the 484 gene associations with a quantitative trait, 355 (83% directionally concordant), 347 (73%) and 210 (74%) were available in SAS, AFR and EAS, respectively. **b**, Of the

80 gene associations with a binary trait, 31 (61% directionally concordant), 31 (61%) and 11 (64%) were available in SAS, AFR and EAS, respectively. Red circles represent associations with $P \leq 0.05$ in the corresponding non-European ancestry. Numbers in the corner of each quadrant represent the proportion of associations in that quadrant, out of the total number of associations in black, and out of the subset with a $P \leq 0.05$ in red. Triangles: associations between binary traits and variants for which the minor allele count (MAC) was 0 in affected individuals.

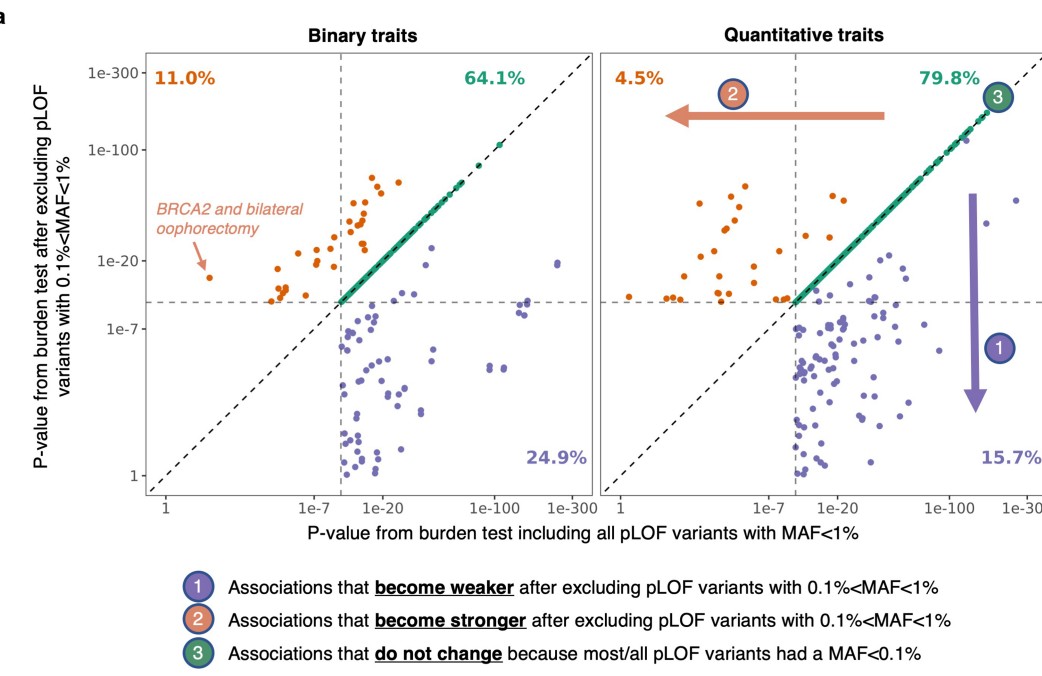

**a**

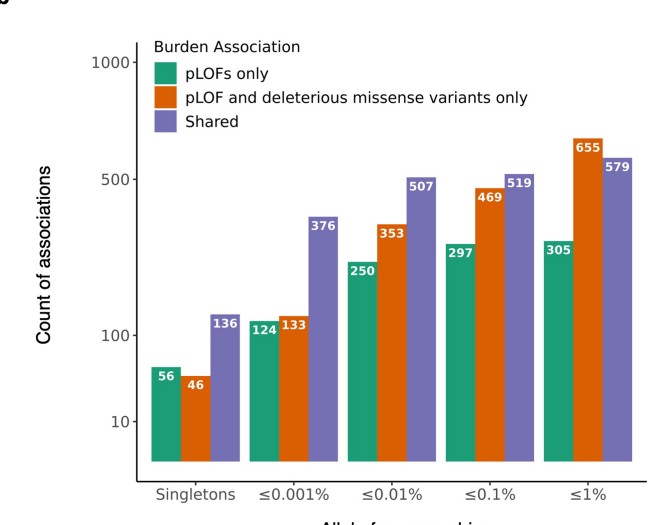

**b**

**Extended Data Fig. 6 | Effect of burden mask composition on yield of significant rare variant associations. a**, Comparison of the trait association *P*-value between burden tests that included pLOF variants with a minor allele frequency (MAF) up to 1% (x-axis) and burden tests that included pLOF variants with a MAF up to 0.1% (y-axis). For a large fraction of associations (64.1% for binary traits, 79.8% for quantitative traits), the association P-value was the same between the 2 burden test strategies, indicating that there were no (or very few) variants with a MAF between 0.1% and 1% included in the burden test.

**b**, Comparison of association yield between burden tests that included pLOF variants only and burden tests that included both pLOF and deleterious missense variants. This comparison was performed separately for the 5 different allele frequency thresholds used to determine which variants were aggregated in the burden test. The proportion of trait associations discovered only when considering both pLOF and deleterious missense variants increased steadily with increasing allele frequency, from 19.3% (46/238) when testing only singletons, to 42.6% (655/1,539) when considering variants with a MAF up to 1%.

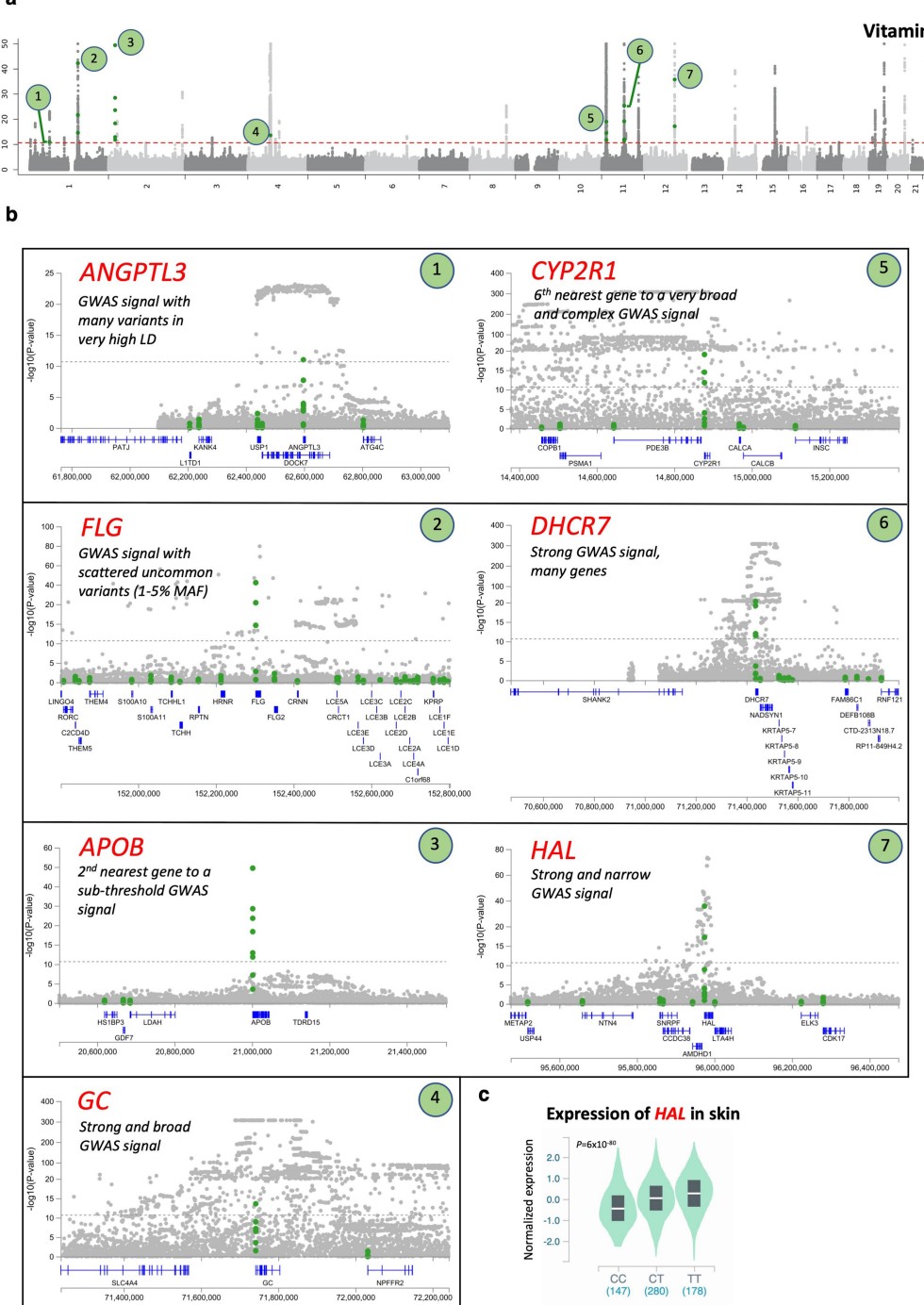

**Extended Data Fig. 7 | Illustration of the utility of exome sequencing data to identify likely effector genes of common variant signals from GWAS.**
**a**, GWAS results for serum vitamin D levels, based on TOPMed imputed data for the same individuals with exome sequencing data. There were 82 independent common variant signals (considering only variants with a MAF>1% and $P < 5 \times 10^{-8}$; of these, 62 were located >10 Mb apart). Analysis of exome sequencing data identified 7 genes with a significant rare variant burden association at $P < 2.18 \times 10^{-11}$ (shown by green circles in the Manhattan plot; up to 10 burden tests performed per gene) after conditioning on GWAS signals. Of these 7 genes (highlighted by the large, green and numbered circles), 5 were the

nearest gene to a GWAS signal: *FLG*, *ANGPTL3*, *GC*, *DHCR7* and *HAL*. P-values were capped at $<10^{-50}$. **b**, Regional association plots are shown for these 7 genes, with green circles showing results from burden tests only, and grey circles showing results from all other variants tested individually, from imputed and exome data. **c**, Association between a sentinel eQTL for *HAL* (rs3819817) and gene expression in skin tissue (sun exposed – lower leg), estimated by GTEx[14] using linear regression. This variant co-localized ($r^2 = 0.97$) with the peak variant associated with vitamin D levels at the *HAL* locus (rs10859995).

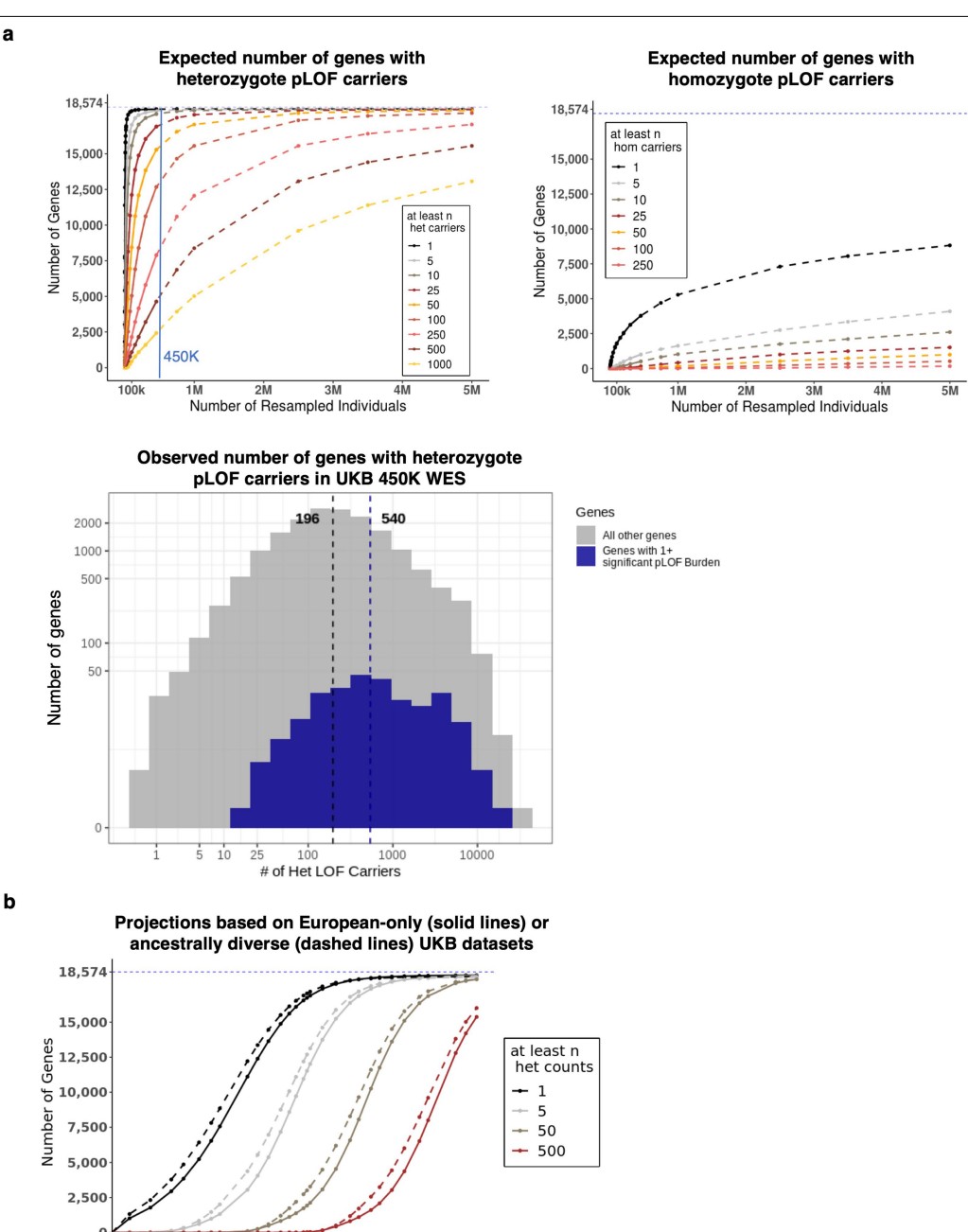

**Extended Data Fig. 8 | Number of genes with pLOF carriers in exome sequencing data. a**, Predicted number of genes with heterozygote (top-left panel) and homozygote (top-right panel) pLOF carriers in exome sequencing data in datasets of up to 5 million individuals. Bottom panel shows distribution of the observed number of heterozygote pLOF carriers per gene in exome sequencing of 454,787 individuals from the UK Biobank. **b**, Predicted number of genes with heterozygote pLOF carriers in 5 million individuals based on a reference dataset of (i) 46K individuals of European ancestry from the UKB (solid lines); and (ii) 46K individuals from the UKB spanning multiple ancestries (dashed lines), including 23K of European ancestry and 23K individuals of other ancestries (10K of South Asian, 9K of African, 2K of East Asian ancestry, 1K of Hispanic or Latin American ancestry and 1K of admixed ancestry).

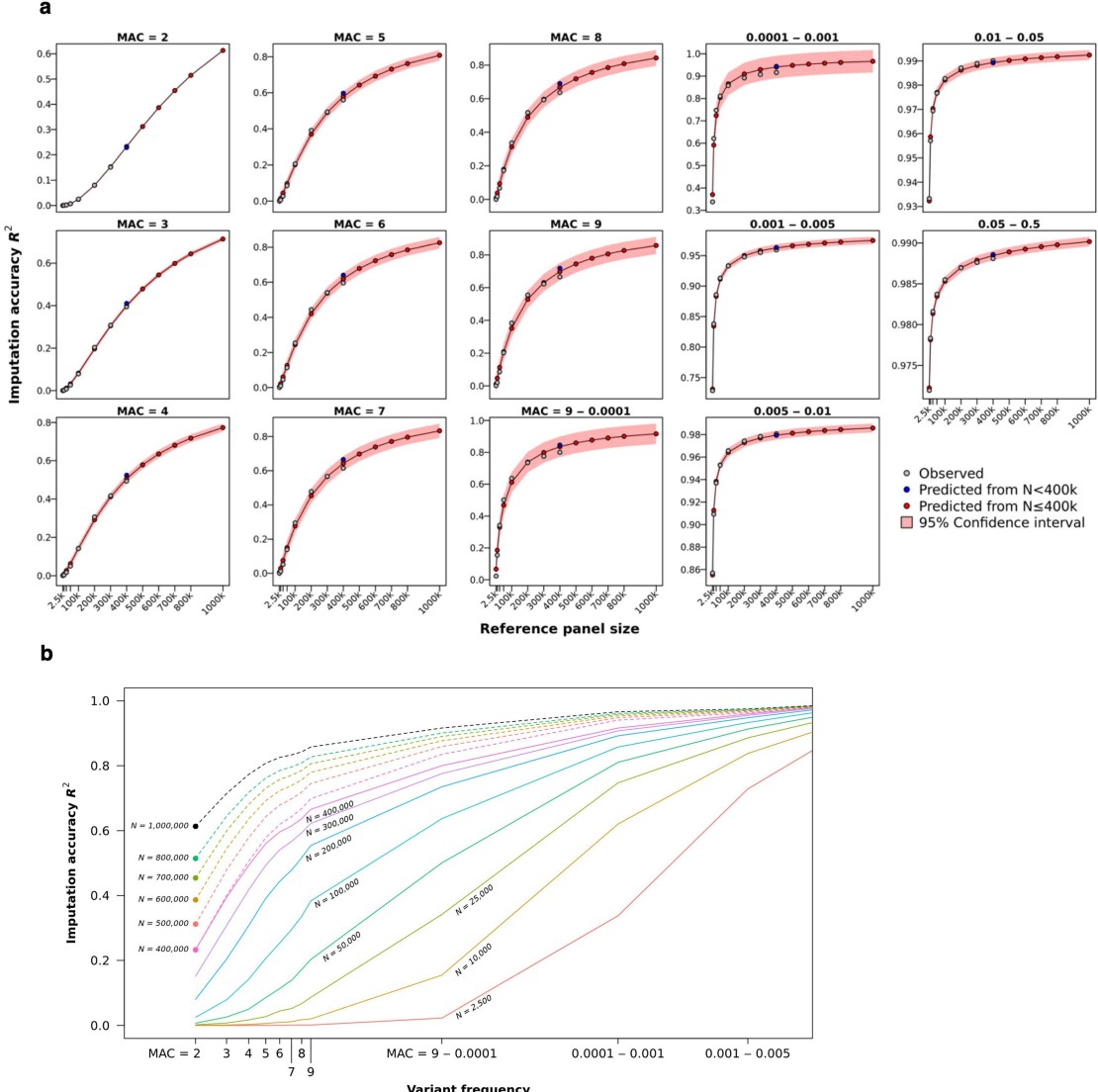

**Extended Data Fig. 9 | Predicted imputation accuracy for variants from exome sequencing as a function of the size of the reference panel using a two-parameter logistic model. a,** Each panel shows the imputation accuracy ($r^2$, y-axis) as a function of the number of individuals included in the reference panel (x-axis), for a given allele frequency bin (estimated in the reference panel). Grey dots show the imputation accuracy that was observed when analysing reference panels with up to 400,000 individuals. Red dots show the imputation accuracy that was predicted for reference panels with >400,000 individuals, obtained by fitting a 2-parameter logistic curve to results from reference panels with ≤400,000 individuals. The fit from this logistic curve is shown by the solid line, with associated 95% confidence intervals shown in light red. The blue dot is the extrapolated value for a reference panel of 400,000 individuals obtained by fitting the curve using only reference panels with <400,000 individuals. **b,** Imputation accuracy ($r^2$, y-axis) is shown as a function of the variant allele frequency (x-axis; minor allele count [MAC] for ultra-rare variants, minor allele frequency [MAF] for variants with MAF>10⁻⁴) and the number of individuals (N) included in the reference panel (different lines). Solid lines show the imputation accuracy that was observed when analysing reference panels with up to 400,000 individuals. Dashed lines show the imputation accuracy that was predicted for reference panels with >400,000 individuals, obtained by fitting a 2-parameter logistic curve to results from reference panels with ≤400,000 individuals.

**Extended Data Table 1 | Novel gene associations identified through the analysis of a burden of singleton variants**

| Gene | Trait | Effect (95% CI) | P-value | N with 0\|1\|2 copies of effect allele[a] | Effect allele frequency |
|---|---|---|---|---|---|
| \multicolumn{6}{c}{Burden of singleton pLOF variants} | | | | | |
| ACAN | Whole-body fat-free mass | -0.74 (-0.91, -0.58) | 2.14E-18 | 423,620\|44\|0 | 5.2E-05 |
| RRBP1 | Apolipoprotein B | -0.83 (-1.02, -0.64) | 3.00E-18 | 410,021\|92\|0 | 1.1E-04 |
| EP400 | Hand grip strength | -0.55 (-0.68, -0.42) | 8.45E-16 | 429,192\|96\|0 | 1.1E-04 |
| CHD2 | Lymphocyte count | 1.16 (0.87, 1.45) | 1.97E-15 | 418,408\|41\|0 | 4.9E-05 |
| SUPT5H | Erythrocyte distribution width | 1.64 (1.23, 2.06) | 7.69E-15 | 419,173\|19\|0 | 2.3E-05 |
| LARP1 | Erythrocyte distribution width | 1.19 (0.88, 1.51) | 1.31E-13 | 419,159\|33\|0 | 3.9E-05 |
| EEF2 | Erythrocyte count | -1.63 (-2.07, -1.18) | 6.51E-13 | 419,181\|12\|0 | 1.4E-05 |
| TNRC6B | Hand grip strength | -0.61 (-0.79, -0.44) | 2.85E-12 | 429,230\|58\|0 | 6.8E-05 |
| HMCN1 | FEV$_1$/FVC (inverted Z-score) | 0.45 (0.32, 0.59) | 1.07E-11 | 343,100\|202\|0 | 2.9E-04 |
| FBN2 | Impedance of arm | 0.45 (0.32, 0.58) | 1.24E-11 | 423,915\|99\|0 | 1.2E-04 |
| \multicolumn{6}{c}{Burden of singleton pLOF and deleterious missense variants} | | | | | |
| CAD | Reticulocyte volume | 0.60 (0.50, 0.70) | 4.51E-31 | 412,190\|311\|0 | 3.8E-04 |
| IGF1R | Leg fat-free mass | -0.41 (-0.49, -0.32) | 1.62E-21 | 423,465\|185\|0 | 2.2E-04 |
| SBNO2 | Lymphocyte count | 0.52 (0.40, 0.64) | 4.24E-17 | 418,221\|228\|0 | 2.7E-04 |
| FGD1[b] | Impedance of arm | -0.47 (-0.60, -0.35) | 7.05E-14 | 423,931\|46\|15 | 9.0E-05 |
| ZNF12 | Insulin growth factor 1 | 0.84 (0.59, 1.08) | 1.86E-11 | 409,874\|52\|0 | 6.3E-05 |

[a]Effect allele for burden tests: individuals were considered to have 0 copies of the effect allele if they were homozygote for the reference allele for all variants included in the burden test; 1 copy of the effect allele if they were heterozygote for at least 1 variant; and 2 copies if they were homozygote for the alternate allele for at least 1 variant.

[b]FGD1 is located on the X chromosome; male hemizygous are included in the number of individuals with 2 copies of the effect allele.

FEV$_1$, forced expiratory volume in 1 second; FVC, forced vital capacity.

# Reporting Summary

## Statistics

For all statistical analyses, confirm that the following items are present in the figure legend, table legend, main text, or Methods section.

| n/a | Confirmed | |
|---|---|---|
| ☐ | ☒ | The exact sample size (*n*) for each experimental group/condition, given as a discrete number and unit of measurement |
| ☐ | ☒ | A statement on whether measurements were taken from distinct samples or whether the same sample was measured repeatedly |
| ☐ | ☒ | The statistical test(s) used AND whether they are one- or two-sided *Only common tests should be described solely by name; describe more complex techniques in the Methods section.* |
| ☐ | ☒ | A description of all covariates tested |
| ☐ | ☒ | A description of any assumptions or corrections, such as tests of normality and adjustment for multiple comparisons |
| ☐ | ☒ | A full description of the statistical parameters including central tendency (e.g. means) or other basic estimates (e.g. regression coefficient) AND variation (e.g. standard deviation) or associated estimates of uncertainty (e.g. confidence intervals) |
| ☐ | ☒ | For null hypothesis testing, the test statistic (e.g. *F*, *t*, *r*) with confidence intervals, effect sizes, degrees of freedom and *P* value noted *Give P values as exact values whenever suitable.* |
| ☒ | ☐ | For Bayesian analysis, information on the choice of priors and Markov chain Monte Carlo settings |
| ☐ | ☒ | For hierarchical and complex designs, identification of the appropriate level for tests and full reporting of outcomes |
| ☐ | ☒ | Estimates of effect sizes (e.g. Cohen's *d*, Pearson's *r*), indicating how they were calculated |

*Our web collection on statistics for biologists contains articles on many of the points above.*

## Software and code

Policy information about availability of computer code

| Data collection | No software was used for data collection. |
|---|---|
| Data analysis | The association analysis package used to perform all genetic associations is available at https://github.com/rgcgithub/regenie. GCTA v1.91.7 was used for approximate conditional analysis. LDSC v1.0.1 was used LD score regression. with SHAPEIT4.2.0 was used for phasing of SNP array data. Imputation was completed with IMPUTE5. |

For manuscripts utilizing custom algorithms or software that are central to the research but not yet described in published literature, software must be made available to editors and reviewers. We strongly encourage code deposition in a community repository (e.g. GitHub). See the Nature Portfolio guidelines for submitting code & software for further information.

## Data

Policy information about availability of data

All manuscripts must include a data availability statement. This statement should provide the following information, where applicable:
- Accession codes, unique identifiers, or web links for publicly available datasets
- A description of any restrictions on data availability
- For clinical datasets or third party data, please ensure that the statement adheres to our policy

Individual-level sequence data have been deposited with UK Biobank and will be freely available to approved researchers, as done with other genetic datasets to date. Individual-level phenotype data are already available to approved researchers for the surveys and health-record datasets from which all our traits are derived. Instructions for access to UK Biobank data is available at https://www.ukbiobank.ac.uk/enable-your-research. Full details for the trait associations with rare variants described in this study are provided in Data S2 and S3. The HapMap3 reference panel was downloaded from ftp://ftp.ncbi.nlm.nih.gov/hapmap/. GnomAD v3.1 VCFs were obtained from https://gnomad.broadinstitute.org/downloads. VCFs for TOPMED Freeze 8 were obtained from dbGaP as described in https://

# Field-specific reporting

Please select the one below that is the best fit for your research. If you are not sure, read the appropriate sections before making your selection.

☒ Life sciences   ☐ Behavioural & social sciences   ☐ Ecological, evolutionary & environmental sciences

For a reference copy of the document with all sections, see nature.com/documents/nr-reporting-summary-flat.pdf

# Life sciences study design

All studies must disclose on these points even when the disclosure is negative.

| | |
|---|---|
| Sample size | Sample size was not predetermined. Association analyses were restricted to the intersection of samples with both exome sequence and array genotypes available after QC. See methods section "Exome sequencing" for details on QC performed. All samples that pass genotype QC and with non-missing phenotype data were included in association analyses. We performed power calculations (Extended data figure 4) that suggest we are well-powered to detect genetic associations under a variety of scenarios, although there may be some traits for which we did not have adequate sample size. |
| Data exclusions | Phenotype selection and QC was performed as described in methods section "Health- and behavior-related phenotypes." Variant level QC was performed as described in methods section "Exome sequencing." Variants with minor allele count less than five were excluded from association testing. The minor allele count threshold was pre-determined based on extensive simulations performed with REGENIE. See https://www.nature.com/articles/s41588-021-00870-7 for additional details. |
| Replication | Replication was attempted for all significant variant-trait associations available for follow-up in the DiscovEHR study. 81% of associations available and powered for replication were confirmed. |
| Randomization | Randomization was not required for the analyses completed in this study. To control for confounding, we performed association analysis with the following covariates included in the regression model: age, age-squared, sex, age-x-sex, 10 ancestry-informative principal components, six exome sequence batch indicator variables, and 20 principal components derived from exome variants with a MAF between 2.6x10-5 and 1%. |
| Blinding | Blinding was not required for the analyses completed in this study. Participant recruitment and phenotype collection were obtained without prior knowledge of sample genotypes. Association analyses were performed with all available samples, without any filtering based on sample genotypes. |

# Reporting for specific materials, systems and methods

We require information from authors about some types of materials, experimental systems and methods used in many studies. Here, indicate whether each material, system or method listed is relevant to your study. If you are not sure if a list item applies to your research, read the appropriate section before selecting a response.

## Materials & experimental systems

| n/a | Involved in the study |
|---|---|
| ☒ | ☐ Antibodies |
| ☒ | ☐ Eukaryotic cell lines |
| ☒ | ☐ Palaeontology and archaeology |
| ☒ | ☐ Animals and other organisms |
| ☐ | ☒ Human research participants |
| ☒ | ☐ Clinical data |
| ☒ | ☐ Dual use research of concern |

## Methods

| n/a | Involved in the study |
|---|---|
| ☒ | ☐ ChIP-seq |
| ☒ | ☐ Flow cytometry |
| ☒ | ☐ MRI-based neuroimaging |

# Human research participants

Policy information about studies involving human research participants

| | |
|---|---|
| Population characteristics | The UK Biobank is a prospective cohort study previously described in detail by Bycroft et al, Nature 2018 (https://www.nature.com/articles/s41586-018-0579-z). Briefly, 94.7% of sequenced participants are of European ancestry, 54.2% are female, the average age at assessment is 58, and the mean BMI is 26. 45% of participants report a history of smoking, and each participant reports 8 inpatient ICD10 3D codes, on average. See supplementary table 1 for additional details. |
| Recruitment | Please see Bycroft et al, Nature 2018. |
| Ethics oversight | Ethical approval for the UK Biobank was previously obtained from the North West Centre for Research Ethics Committee (11/NW/0382). The work described herein was approved by UK Biobank under application number 26041. Approval for |

DiscovEHR analyses was provided by the Geisinger Health System Institutional Review Board under project number 2006-0258. Informed consent was obtained for all study participants.

Note that full information on the approval of the study protocol must also be provided in the manuscript.

