## [Peer Review File · Nature]

Manuscript Title: Exome sequencing and analysis of 454,787 UK Biobank participants

Reviewer Comments & Author Rebuttals

Reviewer Reports on the Initial Version:

Referee #1 (Remarks to the Author):

Authors conducted a whole-exome sequencing (WES) of 450k individuals of the UK biobank. This is the largest WES of the single cohort ever, and provides tremendous resources for the community. They identified >12 million coding variants, which provided a comprehensive catalog of loss-of-function (LOF) and missense variants. A genome-wide association study (>3,500 traits) on single variants and gene-based burden test was conducted for the individuals with European ancestry. This identified risk of the rare variants on a set of clinically important phenotypes, independently from common variants. Associations were further verified in non-European UK biobank individuals and independent cohorts. Of note, this large WES data will be publicly released, which should greatly contribute to the community. This reviewer acknowledges essential value of this manuscript, and has few comments.

1. Definition of the significance threshold of WES-wide association is not fully clear. They set at 2.18×10^{-11} , which suggests around 2.3×10^9 independent tests when $\alpha=0.05$. Please describe composition of this value in the main text.
2. Authors adopted their own burden test. As they described, this is a one-sided test and does not consider accumulation of both susceptible and protective alleles in the same gene contrary to the two-sided tests (e.g., SKAT or SKAT-O). This can be a limitation of this study, since it can miss certain fraction of the rare variant association signals. This reviewer recommends to additionally run the two-sided test.
3. This manuscript focuses on drug target enrichment of the identified risk genes, of which this reviewer feels interests. However, current assessments are on empirical description of respective genes, and lack overall summary on whether WES-associated genes are enriched in targets of currently indicated drugs (especially those indicated for the tested-traits themselves), when relatively compared with total WES target genes.
4. It was not clear how rare SNV and indels separately or coordinately conferred trait associations. Gross summary and introduction of some genic examples are fine. For example, association signals of the ACAN region with anthropometric traits (in Table 1) are suggested to be from not SNV but VNTR.
5. Combinatory analysis with rare and common variants are interesting. How about relationship of distributions between GWAS-derived polygenic risk score (PRS) and rare variants in trait associations in a population scale?
6. In page 12, "Similar results were observed for binary traits": This should be explained more in detail in the main text.
7. Authors assessed independent associations of the rare variants from GWAS-derived common risk variants, which is interesting. On the other hand, how about independent GWAS-derived common variant risk from causal rare variants? Is there any locus where (some proportion of) common variant associations were explained by surrounding rare variants? This should answer the long mystery of "synthetic association" hypothesis, which have not been clearly answered by the previous relatively small-scaled WES studies.

8. This reviewer expects WES data of individuals will be publicly available, but its detail was not clear in description at "Data and code availability section". Can you clearly describe how one can access the individual-level data?

Referee #2 (Remarks to the Author):

This manuscript describes exome sequencing of the full UK Biobank cohort (N=454K) and subsequent association analysis with ~4,000 health-related traits. These analyses reveal a wealth of novel associations likely to be driven by rare protein-altering variants, and the authors highlight several associations implicating genes that may be potential therapeutic targets. The associations appear to be impressively robust, as evidenced by supporting evidence (when available) from replication in the DiscovEHR cohort, consistency across ancestries, consistency across related traits, and/or independent common variant GWAS signals. Additionally, the manuscript provides insights into the empirical behavior of exome burden analyses and the accuracy of large-panel imputation that will inform future analyses.

Overall, the study represents a major advance describing exciting initial results from a data set that is sure to enable many further discoveries in the years to come. The analyses appear to be carefully conducted overall. However, I did have some questions and suggestions for potential improvements:

The authors state that they found "no evidence for a substantial impact of population structure or unmodeled relatedness on the results (fig. S2)." I think Fig. S2 supports this conclusion for the rarer variants and for binary traits, but the lambdaGC values do generally appear to be larger than 1 for quantitative traits (in MAF>1% and 0.1%<MAF<1% categories of variants). This behavior is probably expected from polygenicity, but a clearer demonstration that the quantitative trait associations are not inflated (e.g., via the LDSC attenuation ratio and/or intercept) would be helpful.

Many of the "8,865 significant associations" that the authors report appear to be either mostly or completely redundant based on Data S2. For many genes, different choices of pLOF MAF thresholds produce exactly the same burden "variants" leading to exactly the same result being counted several times. This information is very helpful to report in Data S2, but focusing in the main text on the number of non-redundant associations (defined in any reasonable way) would make more sense.

In the analysis of protective associations with disease traits at the "more liberal" significance threshold of $P < 1e-7$, I was initially surprised that all associations identified at this threshold appear to be genuine statistical associations, with not even one false positive that reached $P < 1e-7$ by chance. I wonder whether the multiple test burden was actually much smaller than (# variants) x (# traits) because for most variant-trait pairs tested, it was impossible for a protective association to reach $P < 1e-7$ because the variant and/or trait were too rare -- i.e., the authors effectively only tested low-frequency variants against fairly common binary traits -- such that $P < 1e-7$ might actually be a sufficiently conservative threshold. All four associations the authors highlight have this property: >5,000 carriers each for SLC9A3R2, SLC27A3, PIEZO1, MAP3K15, all associated with common binary traits (hypertension, asthma, varicose veins, T2D).

Given that all of the above associations involved >5,000 carriers, was WES data actually necessary to find these associations? Could they all have been discovered from the TOPMed-imputed genotypes?

Does the suggestion that "blocking SLC9A3R2 could provide an attractive means for managing blood pressure" (line 150) make sense given that the effect on SBP / DBP was only -1.85 / -1.01

mmHg in carriers? Is there any evidence of a larger effect in homozygotes / compound hets?

For the SLC27A3-asthma result, the consistency with eosinophil counts and replication in DiscovEHR are convincing, but I was less sure about the the independent common missense variant rs34527123 (CADD=18.3, benign/tolerated according to PolyPhen2/SIFT). Does statistical fine-mapping support causality of this variant, or could it simply be tagging some other common variant (rather than being a gain-of-function variant, as the authors suggest)?

The MAP3K15 variant is reported to have a P-value of $2.3e-6$ for T2D (line 219) but was mentioned in the list of protective associations reaching $P < 1e-7$. Was $2.3e-6$ a typo or did it reach $P < 1e-7$ for another trait? Also, the gene name seems to have a typo in the Abstract ("MAP5K15").

In the cross-ancestry analyses, the authors report that "Similar results were observed for binary traits (fig. S9b)" (line 243) but the concordance in Fig. S9b appears to be somewhat lower than for the quantitative traits.

The section describing the impact of burden test composition on the yield of genetic associations is interesting. The authors observe that aggregating variants across a wider range of allele frequencies generally appears more powerful (p. 13). However, are an increasing fraction of the associations potentially explained by LD with common variants as the MAF threshold rises? For example, what fraction of the $971 - 804 = 167$ burden associations discovered only at $MAF < 1\%$ lost significance upon conditioning on GWAS sentinel variants? In a similar spirit, are the 804 of 1,775 associations discovered exclusively by aggregating pLOF and deleterious missense variants enriched for associations that might tag common variants?

I couldn't seem to find a row for the association between ANGPTL3 and Vitamin D (highlighted on p. 16) in Data S2. Did this signal survive conditioning on GWAS sentinel variants?

I am unsure whether the HAL eQTL should be considered a "gain-of-function" variant (line 359) if it only alters expression.

The imputation results (Fig. 4) are probably somewhat over-optimistic in that they represent the ideal scenario of imputation using a reference panel with perfectly-matched ancestry: i.e., imputation into another UK cohort would achieve the indicated accuracy, but imputation accuracy even into European-ancestry cohorts outside the UK would presumably be lower. More generalizable results could be obtained by imputing into an external data set such as 1000 Genomes (aggregating variants within MAF ranges to obtain sufficient sample sizes).

Referee #3 (Remarks to the Author):

This is an important manuscript from the Regeneron team summarizing the results of phenome-wide rare variant association tests on the largest exome sequence dataset analyzed to date comprising of nearly 0.5 million UK Biobank participants. Beyond large sample size and comprehensive analyses, the study has numerous strengths including high quality of sequence data, unbiased and computationally efficient methodology to identify rare variant associations, high level of statistical stringency used to define significant associations, incorporation of GWAS results with control for common variant signals in rare variant association tests, and systematic replication studies in independent exome datasets. In several ways, this study is setting up standards for future large-scale rare variant association studies.

I would like the authors to address few relatively minor issues that require further clarification or expanded discussion:

1. Replication of RV associations in African, South Asian and East Asian ancestries: please state

clearly how many of the 8,865 associations identified in the European cohort replicate with a direction-consistent effect AND $P < 0.05$ in each ancestral group.

2. Replication of RV associations in the Geisinger DiscovEHR cohort: the current presentation of these results is confusing and it is not clear why only the 279 “strongest trait-variant associations” are being tested for replication. Please state clearly how many of the 8,865 associations identified in the UKBB cohort have a matching phenotype in the Geisinger cohort, and how many of these associations replicate with a direction-consistent effect AND $P < 0.05$ (similar to #2 above).

3. Phenotype definitions and association analyses: please provide more information on the frequency distribution of the tested binary phenotypes; how many phenotypes can be considered as rare diseases vs. more common traits? Also, please clarify how sex chromosomes and sex-specific phenotypes (e.g. prostate cancer, breast cancer etc.) are being handled in the analysis.

4. Overlap between CV signals from GWAS and RV signals: please provide more detail on the approaches used for GWAS analyses including methods used for association testing (was REGENIE used for GWAS?), PCA adjustments (same as for RV analyses?), genomic inflation factors (not provided). Also, the $P < 10^{-7}$ to define independently associated signals by GCTA-COJO for inclusion in the conditional analyses of RV associations may need a better justification -- how was this threshold selected?

5. Beyond 500,000 exomes: Please clarify in the results if the projections of pLOF carriers for a dataset of 5 million sequenced individuals are based on the UKBB Europeans, or all ancestries included in the UKBB. Given that UKBB is dominated by Europeans, would similar projections be expected for sequencing studies of more diverse populations?

6. Whole-genome sequencing and imputation: The analysis of minimal imputable MAF by varying size of the reference panel is very helpful, but again this analysis is based predominantly on UKBB Europeans. The authors should comment on the relevance of these findings to the imputation of non-European genomes.

7. Results on Page 16 line 343: “Rare pLOFs and deleterious missense variants in ANGPTL3 ... were associated with higher vitamin D levels, suggesting a role for ANGPTL3 in vitamin D production in the kidney.” Vitamin D is not produced in the kidney, but rather activated by 1-OH hydroxylation. As far as I know, the conclusion that ANGPTL3 would be involved in the kidney activation of vitamin D is not based on evidence - in fact, liver is a much more plausible causal tissue given that 25-OH vitamin D is the major circulating form rather than 1,25-(OH)₂. Notably, 25-OH hydroxylation of vitamin D occurs in the liver and liver is the predominant site of ANGPTL3 expression. Additionally, on page 16 line 346: “The second gene that we implicate for the first time in vitamin D biology was HAL...” For clarity, it would be helpful to mention in this paragraph that the first step of vitamin D synthesis takes place in the skin and involves UV light.

8. The discussion section should refer directly to study limitations, which are presently buried in the supplementary text. The following should also be considered/added to the limitations: 1) defining UKBB phenotypes based on electronic health records/ICD codes or self-report may result in phenotype misclassification and/or missing data limiting the power to detect associations for some phenotypes; 2) there is still limited power to detect associations for rare phenotypes -- it would be helpful to get a better idea how many of the analyzed phenotypes are relatively rare (e.g. $< 0.01\%$, $< 0.1\%$, $< 1\%$ frequency) vs. relatively common (e.g. $> 5\%$, $> 10\%$, $> 20\%$ frequency) and comment on the power implications; 3) under-representation of non-European participants in the UKBB may limit the accuracy of projections for genome imputations and/or expected RV counts when expanding sequencing to more diverse populations. It would also be helpful to state as one of the limitations that the assessment of variant segregation or de novo occurrence was not possible because of the population-based design of the UKBB.

(Minor) Figure 1. please provide a more detailed legend for the HSPG2 and alkaline phosphatase example (panel C). Are the presented single variant statistics (grey) for RVs only or for RVs+CVs? I assume RVs, but please state clearly in the legend. Also the presentation of burden p-values as dots is confusing, would specify that green indicates a burden test P-value for improved clarity.

Author Rebuttals to Initial Comments:

Referee #1

1. Definition of the significance threshold of WES-wide association is not fully clear. They set at 2.18×10^{-11} , which suggests around 2.3×10^9 independent tests when $\alpha=0.05$. Please describe composition of this value in the main text.

On page 4 of the main text, we had the following sentence describing how we arrived at a $P=2.18 \times 10^{-11}$:

“Overall, we performed a total of ~2.3 billion association tests (table S5), with no evidence for a substantial impact of population structure or unmodeled relatedness on the results (fig. S2). We found 8,865 significant associations – involving 564 genes and 492 traits (fig. S3) – at a $P \leq 2.18 \times 10^{-11}$, which corresponds to a Bonferroni correction for multiple testing (Data S2; <0.05 association signals expected across all tests).”

To make this point clearer, we have revised the second sentence in this section to instead read:

“We found 8,865 significant associations – involving 564 genes and 492 traits (Extended Data Figure 3) – at a $P \leq 2.18 \times 10^{-11}$, which corresponds to a Bonferroni correction for multiple testing (i.e. $P \leq 0.05 / 2.3$ billion tests; at this threshold, <0.05 association signals expected by chance across all analysis results).”

2. Authors adopted their own burden test. As they described, this is a one-sided test and does not consider accumulation of both susceptible and protective alleles in the same gene contrary to the two-sided tests (e.g., SKAT or SKAT-O). This can be a limitation of this study, since it can miss certain fraction of the rare variant association signals. This reviewer recommends to additionally run the two-sided test.

We fully agree that approaches such as SKAT or SKAT-O will identify additional association signals, particularly when coding variants in the same gene act in opposite directions, as previously

documented (for example, see PMID 22863193). As such, we can safely predict that applying approaches such as SKAT to the UKB 450K WES data will discover additional associations. We now explicitly address this limitation in the discussion and continue to note that the data, which are all available to the scientific community, should enable additional follow-up analyses and discoveries.

One reason to focus on burden tests that assume all variants act in the same direction is that they facilitate interpretation and identification of attractive targets for therapeutic inhibition or activation. In addition, we have extensive experience with our burden tests (see, for example, PMIDs 33087929, 34115965, 34210852) that they yield results that are robust, straightforward to meta-analyze and to compare between studies, and have well controlled type 1 error, as evidenced by genomic inflation factors close to (or less than) 1.

Therefore, although we do agree that running SKAT across all traits is interesting and worthwhile, this would be a substantial undertaking at this point, with nearly 4K traits to test in UKB, plus replication in GHS. On the flip side, not performing such analyses leaves the door open for other groups to make new discoveries using the UKB 450K WES data.

3. This manuscript focuses on drug target enrichment of the identified risk genes, of which this reviewer feels interests. However, current assessments are on empirical description of respective genes, and lack overall summary on whether WES-associated genes are enriched in targets of currently indicated drugs (especially those indicated for the tested-traits themselves), when relatively compared with total WES target genes.

In recent years, several studies have shown that genes associated with trait variation in humans are enriched among approved drug targets (Nelson et al, King et al, Minikel et al). We did not tackle this question explicitly in our original submission but, following the reviewer's request, have now specifically tested if WES-associated genes are enriched among drug targets when compared to all other genes tested. To this end, we (i) obtained from DrugBank a list of 385 genes that are the target of FDA-approved drugs (PMID 29126136; https://github.com/macarthurlab/gene_lists/blob/master/lists/fda_approved_drug_targets.tsv), of which 381 were among the 18,811 genes with WES association results in our data; and (ii) tested if the 564 genes associated with at least one trait in our analysis of the UKB 450K were enriched among that list of 381 FDA target genes.

As shown in table below, we found that 36 of 564 (6.4%) WES-associated genes encode FDA-approved drug targets, when compared to 345 out of the remaining 18,317 (1.9%) genes tested, a 3.6-fold enrichment (Fisher's exact test $P=1.7 \times 10^{-9}$).

		Associated with at least one trait?		
		No	Yes	Total
FDA-approved drug target?	No	17,972	528	18,500
	Yes	345	36	381
	Total	18,317	564	18,811

We have now added results from this analysis to the main text (page 5) and annotated these 36 genes in Supp Table 6. We also added a paragraph to the caveats section (in the Supplementary Discussion) acknowledging two main limitations of this analysis: (i) the traits associated in the WES data with each of the 36 genes that are FDA-approved targets may be unrelated to the diseases for which the corresponding drug is approved for; and (ii) for some of the remaining 345 genes that are FDA-approved drug targets, we may not have tested any trait that is related to the approved indication.

4. *It was not clear how rare SNV and indels separately or coordinately conferred trait associations. Gross summary and introduction of some genic examples are fine. For example, association signals of the ACAN region with anthropometric traits (in Table 1) are suggested to be from not SNV but VNTR.*

Of the 564 lead gene-trait associations listed in Supp Table 6, 149 were with an individual rare variant (i.e. not a burden of rare variants). Of these, for 20 genes (3.5% of the 564 total) the most significant association was with an indel. For example, 1:84570692:G:GT in *CTBS* was associated with peak expiratory flow; 2:237388132:TC:T in *COL6A3* was associated with corneal resistance factor. For the other 129 genes (=149-20; 22.9% of the 564 total), the lead trait association was with an SNV.

For the remaining 415 genes (564-149), the lead trait association was with a burden test. To help understand if these burden associations were primarily driven by SNVs or indels, we examined results for individual variants included in the burden test (we specifically considered SNVs and indels with $MAC \geq 5$; variants with a MAC of 1-4 were not tested individually). We found that:

- 1) 14 genes (2.5% of the 564 total) had at least one indel with a $P \leq 10^{-7}$ but no SNV with a $P \leq 10^{-7}$. Examples include the association between *ANO5* and aspartate aminotransferase (burden test likely driven by 11:22221100:C:CA); *CXCL6* and neutrophil count (4:73837092:G:GT); *PKD2* and cystic kidney disease (4:88065406:G:GA).
- 2) 48 genes (8.5% of the 564 total) had no indel with a $P \leq 1e-7$ but at least one SNV with a $P \leq 1e-7$.
- 3) 3 genes (0.5% of the 564 total) had at least one indel with a $P \leq 10^{-7}$ and at least one SNV with a $P \leq 10^{-7}$.

All other genes (415-14-48-3=350) did not have any individual variant (indel or SNV) with a $P < 10^{-7}$, and so for these it is not clear if the associations are driven by indels, SNVs or a combination of both.

Therefore, the lead trait associations were primarily driven by an individual indel for 34 genes (20+14; 6.0% of total); by an individual SNV for 177 genes (129+48; 31.4%); and by a combination of both an individual indel and an individual SNV for 3 genes (0.5%).

To incorporate this new information into the revised manuscript, we added two additional columns to Supp Table 6, respectively indicating:

- 1) If the observed burden association was primarily driven by an individual SNV, indel or both; and
- 2) The ID for the individual variant that was the primary driver of the burden association.

5. Combinatory analysis with rare and common variants are interesting. How about relationship of distributions between GWAS-derived polygenic risk score (PRS) and rare variants in trait associations in a population scale?

This is definitely an interesting idea, which we explored recently in the context of a paper that performed GWAS and exome-wide analyses of body mass index (see Figure 6 in PMID 34210852). However, we feel that this type of analysis is beyond the scope of (and would significantly lengthen) the current manuscript, being a better fit for follow-up studies that focus on specific traits and/or genes.

6. In page 12, "Similar results were observed for binary traits": This should be explained more in detail in the main text.

Done as requested. The proportion of traits for which there were concordant effects between European and non-European ancestries was slightly lower for binary traits (61% to 64%) when compared to quantitative traits (73% to 83%), likely due to lower power for the former. We now describe this in more detail the main text. We have also focused this section on a non-redundant set of 564 associations, as requested by reviewer #2 (see comment #7). The relevant section now reads:

"When we focused on the 564 non-redundant associations (i.e. strongest association per gene, 484 with a quantitative trait, 80 with a binary trait; Supplementary Table 6), we found that a large fraction of associations was shared across ancestries for quantitative traits but less so for binary traits, likely

due to low power. Specifically, for quantitative traits, effect sizes were directionally concordant for 83% of associations in individuals of SAS, 73% of AFR and 74% of EAS ancestry, increasing to >90% when considering associations with a $P \leq 0.05$ (Extended Data Figure 7a). For binary traits, consistent effects were observed for 61% of associations in SAS, 61% in AFR and 64% in EAS (Extended Data Figure 7b). Similar results were observed when assessing directional consistency across the full set of 8,865 associations (Supplementary Figure 5).”

7. Authors assessed independent associations of the rare variants from GWAS-derived common risk variants, which is interesting. On the other hand, how about independent GWAS-derived common variant risk from causal rare variants? Is there any locus where (some proportion of) common variant associations were explained by surrounding rare variants? This should answer the long mystery of “synthetic association” hypothesis, which have not been clearly answered by the previous relatively small-scaled WES studies.

The current literature seems clear (PMID 21267061) that if synthetic associations are prevalent then they would tend to overlap with loci identified from well powered linkage studies, as multiple large effect causal variants would be very amenable to detection by non-parametric linkage analysis. This is not something that has been widely reported in the GWAS field.

In our analysis, when common and rare variant (RV) signals were located close together, the common variant (CV) signal was nearly always several orders of magnitude more significant (even if associated with a smaller allelic effect or odds ratio), and thus could not be explained through synthetic association. For example, across the 584 examples of a RV near a CV signal shown in Supp Table 18, the median $-\log_{10}(P\text{-value})$ for the CV signal was 57, whereas for the nearby RV signal it was 20. This strongly argues against CV signals being explained by RV signals.

Nonetheless, to examine this question thoroughly would require an extensive extra analysis, which we believe is beyond the scope of the current paper. There will be many rare variants in many flanking genes of each common variant, and the details of which variants to condition on, or whether to condition on the masks themselves, would need investigation.

8. This reviewer expects WES data of individuals will be publicly available, but its detail was not clear in description at “Data and code availability section”. Can you clearly describe how one can access the individual-level data?

Thank you for pointing out that we had not clarified how the individual-level data could be accessed. All primary data has been deposited with UK Biobank and will be available to the community. This

includes all the phenotype data (health records, surveys, imaging data, etc.) which are already available to approved UK Biobank researchers. We provided UK Biobank our primary exome sequence data at the beginning of January 2021, and they are committed to making these data available to the community around September 2021. This will constitute the largest single set of sequenced humans and phenotypes available to the scientific community. The access mechanisms are fair and straightforward and there are thousands of approved users and publications for previous tranches of data.

We have added the following sentence to the “Data and code availability section:

“Individual-level sequence data have been deposited with UK Biobank and will be freely available to approved researchers, as done with other genetic datasets to date. Individual-level phenotype data are already available to approved researchers for the surveys and health-record datasets from which all our traits are derived.”

Referee #2

1. The authors state that they found "no evidence for a substantial impact of population structure or unmodeled relatedness on the results (fig. S2)." I think Fig. S2 supports this conclusion for the rarer variants and for binary traits, but the lambdaGC values do generally appear to be larger than 1 for quantitative traits (in MAF>1% and 0.1%<MAF<1% categories of variants). This behavior is probably expected from polygenicity, but a clearer demonstration that the quantitative trait associations are not inflated (e.g., via the LDSC attenuation ratio and/or intercept) would be helpful.

To address the concern that common variant associations with quantitative traits might be inflated, we added to Supp Data 1 (which originally simply listed all 3,998 traits tested) and Supp Figure the following additional information:

1. The genomic inflation factors from the exome-wide association analysis for all 3,998 traits, split by MAF bin. These are the same data that we showed graphically in Extended Data Figure 2.
2. The genomic inflation factors from the TOPMed GWAS for the 492 traits that had at least one RV association in the exome data, estimated based on variants with a MAF>1%. This was also requested by reviewer 3. We did not run TOPMed GWAS for the 3,506 traits (=3,998-492) that had no RV associations.
3. LDSC attenuation ratio and intercept from the TOPMed GWAS for the 492 traits that had at least one RV association in the exome data. We did not run LDSC on common variants from the exome data, given the very limited overlap (30K SNPs) between common coding variants from exome sequencing and the HapMap3 backbone used to estimate LD scores with LDSC.

These new data show that (i) genomic inflation factors for common variants are very similar between the exome-wide and GWAS analyses, as expected (left scatterplot below); and (ii) the LDSC attenuation ratios and intercepts (center histogram and right scatterplot) are consistent with no substantial impact of population structure or unmodeled relatedness on the results from common variants, as we have shown previously when we first applied REGENIE to UK Biobank data (PMID 34017140).

2. Many of the "8,865 significant associations" that the authors report appear to be either mostly or completely redundant based on Data S2. For many genes, different choices of pLOF MAF thresholds produce exactly the same burden "variants" leading to exactly the same result being counted several times. This information is very helpful to report in Data S2, but focusing in the main text on the number of non-redundant associations (defined in any reasonable way) would make more sense.

There is indeed considerable redundancy among the top 8,865 associations. This redundancy arises because we tested multiple (often correlated) variants/burden tests per gene, and multiple (often correlated) traits per variant/burden test. For presentation purposes, we had to consider (i) how to remove this redundancy in an intuitive way; and (ii) when to present results before vs. after removing redundant associations.

We removed redundancy by reducing the 8,865 associations to only the most significant association per gene, that is 564 associations (specifically, the most significant trait-variant pair per gene). These non-redundant associations are presented in Supplementary Table 6 and are the focus of most sections of the manuscript. For example, we specifically used this non-redundant set of associations to assess replication in the DiscovEHR cohort. This was important to facilitate interpretation of the replication results (e.g. to be able to quantify the number of significant associations expected by chance alone due to multiple testing). For a small subset of analyses, it did not seem appropriate to us to focus only on the non-redundant set of associations. For example, on pages 5 to 11, where we describe specific examples of noteworthy associations, we often present associations with multiple correlated variants and traits per gene (e.g. association between *MAP3K15* and serum glucose, hemoglobin A1c and type-2 diabetes), as this helps characterize each gene association in more detail.

Thus, we feel that we need to strike the right balance between presenting all results vs. non-redundant results, without confusing the reader. We suggest the following compromise:

1) We acknowledge in the text (page 5) that there is a lot of redundancy among the 8,865 associations (as discussed above) and make available two sets of filtered association results as follows:

a) Results for which we removed redundancy due to having tested multiple variants/burden tests per gene. This corresponds to retaining only the most significant association per gene per trait. When we did this, the 8,865 associations reduced down to 2,283 associations (an average of 4 associated traits per gene). We have now included an additional column in Supp Data 2 that can be used to filter results down to this set of 2,283 associations.

b) Results for which we removed redundancy due to having tested multiple variants/burden tests per gene **and** multiple traits per variant/burden test. This corresponds to retaining only the most significant association per gene. These 564 associations are provided in Supp Table 6 (as described above) and can also be easily identified in Supp Data 2. We now explicitly indicate in the main text (page 5) that Supp Table 6 contains this set of non-redundant associations.

2) In each section, we focus the analysis on either the full set of results or the non-redundant results, as appropriate for that analysis. We went through each section and flagged only one instance where in our view it could have been more intuitive to use the non-redundant set of results: the section that investigated the extent to which associations identified in Europeans were consistent across other ancestries. As we did for replication in the DiscovEHR cohort, it is appropriate (and arguably more consistent) to compare results only using the set of 564 non-redundant associations (one per gene). As such, we have updated that section and Extended Data Figure 7 accordingly. For completeness, we also kept the original comparison of effect sizes (which were based on the full set of 8,865 associations) in Supp Figure 6.

3. In the analysis of protective associations with disease traits at the "more liberal" significance threshold of $P < 1e-7$, I was initially surprised that all associations identified at this threshold appear to be genuine statistical associations, with not even one false positive that reached $P < 1e-7$ by chance. I wonder whether the multiple test burden was actually much smaller than $(\# \text{ variants}) \times (\# \text{ traits})$ because for most variant-trait pairs tested, it was impossible for a protective association to reach $P < 1e-7$ because the variant and/or trait were too rare -- i.e., the authors effectively only tested low-frequency

variants against fairly common binary traits -- such that $P < 1e-7$ might actually be a sufficiently conservative threshold. All four associations the authors highlight have this property: >5,000 carriers each for *SLC9A3R2*, *SLC27A3*, *PIEZO1*, *MAP3K15*, all associated with common binary traits (hypertension, asthma, varicose veins, T2D). Given that the all of the above associations involved >5,000 carriers, was WES data actually necessary to find these associations? Could they all have been discovered from the TOPMed-imputed genotypes?

Given the large number of carriers supporting the four associations the reviewer alludes to (*SLC9A3R2*, *SLC27A3*, *PIEZO1* and *MAP3K15*), it is reasonable to hypothesize that we might have detected these associations using TOPMed-imputed genotypes.

To test that possibility, for each gene, we first asked if any individual pLOF or deleterious missense variant was (i) well imputed in the UKB TOPMed data; and (ii) had a $P < 10^{-7}$ in the TOPMed GWAS for the corresponding trait. We found this was the case for two associations: *PIEZO1* and varicose veins - Pro2510Leu (16:88715642:G:A), imputation info score = 0.97, OR=0.69, $P = 3 \times 10^{-8}$ (7,447 carriers); *SLC27A3R2* and hypertension - Arg171Trp (16:2036420:C:T), imputation info score = 0.99, OR=0.81, $P = 4 \times 10^{-10}$ (5,168 carriers). So the associations with *PIEZO1* and *SLC27A3R2* can definitely be discovered through imputation and this is now stated in the text.

For the remaining two genes, we asked if there was a significant association ($P < 10^{-7}$) between a burden of pLOFs/deleterious missense variants and the corresponding traits using TOPMed-imputed (rather than exome sequencing) data. In table below we show results for the burden test that identified these associations in the exome data (which included pLOF + deleterious missense variants with MAF < 1%) but now considering all variants available in TOPMed, irrespective of imputation info score. Almost identical results were observed when we considered only variants imputed with high accuracy (eg. info score > 0.6 or > 0.9; not shown). We found no burden associations at $P < 10^{-7}$ with either gene, but we note consistent associations at $P < 10^{-4}$ for both:

Gene	Trait	Effect (95% CI)	P-value	Cases	Controls
MAP3K15	Diabetes	0.78 (0.71,0.86)	1.19E-06	21863 110 85	402297 3619 1765
SLC27A3	Childhood asthma	0.66 (0.54,0.81)	4.98E-05	16864 88 0	278247 2225 1

These results show that only two of the associations we report with these four genes are discoverable at $P < 10^{-7}$ with TOPMed imputed data at current sample sizes (*PIEZO1* and *SLC9A3R2* associations). For the other two genes (*MAP3K15* and *SLC27A3*), some of the pLOF/deleterious missense variants driving the burden test associations were either not available in the TOPMed data, were too rare to be imputed adequately, or were drowned out by poor imputation results at other pLOF/deleterious variants. For example, for *SLC27A3*, 314 variants were included in the burden test using exome sequencing data, whereas only 124 were present in the TOPMed data, including 72 imputed with an info score > 0.3.

Gene	N variants in burden test	
	Exome sequencing data	TOPMed data (All INFO >0.3 INFO >0.6 INFO >0.9)
MAP3K15	369	17 17 17 17
SLC27A3	314	124 72 50 17

We have added this information to page 9 and to new Supp Tables 9 and 10.

4. Does the suggestion that "blocking SLC9A3R2 could provide an attractive means for managing blood pressure" (line 150) make sense given that the effect on SBP / DBP was only -1.85 / -1.01 mmHg in carriers? Is there any evidence of a larger effect in homozygotes / compound hets?

This is a great question. The ability of a genetic effect to translate into a clinically meaningful drug effect depends on many factors, of which we highlight two that are relevant here. First, the extent to which the variants that we tested in SLC9A3R2 (pLOFs and deleterious missense variants) impact gene function. For example, do pLOFs and deleterious missense variants have a comparable effect on gene function? Or do deleterious missense variants cause only partial loss of function? Second, as the reviewer points out, is the effect in homozygote carriers greater than in heterozygote carriers? Homozygote carriers of pLOFs potentially have no functional copy of the gene and therefore more closely mimic the effect of a drug that results in 100% inhibition, when compared to heterozygote carriers (who potentially have one functional copy of the gene). Regarding these two points, we note the following:

- 1) pLOF vs deleterious missense. In the heterozygote state, a burden of pLOF+deleterious missense variants was associated with a reduction in SBP of 1.85 mmHg. But a burden of pLOFs alone had a ~2.5x larger effect, being associated with a reduction in SBP of 4.60 mmHg (mean SBP=133.4 in heterozygous carriers [n=57] vs. mean SBP=138.1 in non-carriers [n=408,154]). Although this difference was not statistically significant (P=0.09), likely due to small sample size, it is consistent with the possibility that pLOF variants have a larger effect on SBP when compared to deleterious missense variants.
- 2) Homozygote vs heterozygote effect. In the homozygote state, a burden of pLOF+deleterious missense variants was associated with a reduction in SBP of 3.40 mmHg: mean SBP=134.7 in homozygote carriers (n=15) vs. mean SBP=138.1 in non-carriers (n= 401,874). This effect is ~2-fold larger than observed in heterozygous carriers, and so it is consistent with an additive effect.

Lastly, we also note that heterozygote carriers of rare pLOF or deleterious missense variants had a 19% lower risk of hypertension (see Supp Table 7). This is a substantial risk reduction.

Therefore, overall, we do believe that the statement “blocking SLC9A3R2” is justified, but have added to this sentence that “functional studies that address this possibility are warranted”.

5. For the SLC27A3-asthma result, the consistency with eosinophil counts and replication in DiscovEHR are convincing, but I was less sure about the the independent common missense variant rs34527123 (CADD=18.3, benign/tolerated according to PolyPhen2/SIFT). Does statistical fine-mapping support causality of this variant, or could it simply be tagging some other common variant (rather than being a gain-of-function variant, as the authors suggest)?

This is also a great question that we had not addressed in our original submission. We think it’s unlikely that the association between the common missense variant rs34527123 in SLC27A3 and childhood asthma (MAF=3%, OR=1.08, P=0.008) or eosinophil counts (beta=0.023, P=1.2x10⁻⁵) is simply tagging the association with some other common variant. This is because:

- 1) The association with rs34527123 remained unchanged (OR=1.08, P=0.008) after we conditioned on the three independent GWAS signals for asthma at this locus (identified with GCTA-cojo). Similarly, there is no GWAS signal with P<10⁻⁷ for eosinophils within 2 Mb of rs34527123. These observations rule out the possibility that the associations with rs34527123 are explained by LD with strong GWAS signals.
- 2) To address the possibility that the association between rs34527123 and asthma is explained by a sub-threshold GWAS signal, we used FINEMAP as suggested to identify a credible set of causal variants at this locus in the UKB TOPMed GWAS. FINEMAP identified five independent signals for asthma at this locus (see table below). Of note, variant rs34527123 is one of 682 variants that are part of signal #5, but this signal is very modest, and so results from FINEMAP need to be interpreted with caution. With this caveat in mind, these results show that the association between asthma and rs34527123 is not explained by LD with a sub-threshold GWAS signal (like signal #4) and lend some support to the possibility that rs34527123 is a causal variant, although larger studies are required to confirm this possibility.

Signal	Lead variant in signal	P-value in asthma GWAS	MAF	Posterior probability of lead variant being causal	N variants part of this signal
1	rs61816766	7.60E-46	0.03	1.000	1

2	rs12731336	1.20E-42	0.04	1.000	1
3	rs18458744	2.40E-29	0.02	0.947	2
4	rs72700915	1.80E-05	0.01	0.541	2
5	rs14388206	0.084	0.003	0.690	682

Despite the observations above, we agree with the reviewer that the association with the common missense variant does not provide unambiguous support for the association with the rare pLOF/deleterious missense variants, and therefore it is not critical for this section. As such, given space constraints (which would make it a challenge to succinctly explain the analyses above), we have opted for removing the association with the common missense variant from this section.

6. The MAP3K15 variant is reported to have a P-value of 2.3e-6 for T2D (line 219) but was mentioned in the list of protective associations reaching $P < 1e-7$. Was 2.3e-6 a typo or did it reach $P < 1e-7$ for another trait? Also, the gene name seems to have a typo in the Abstract ("MAP5K15").

Apologies for this confusion. There a couple of explanations for this apparent inconsistency.

First, MAP3K15 was associated with two diabetes-related phenotypes:

- “Doctor-diagnosed diabetes (2443)” (22K cases, 407K controls), defined solely based on responses to the question “Has a doctor ever told you that you have diabetes?”, which was included in the UKB touchscreen questionnaire. This phenotype had the most significant association with MAP3K15 (OR=0.80, $P=8.57 \times 10^{-8}$) and so it is the phenotype that is flagged in the section that describes protective associations with disease outcomes at $P < 10^{-7}$ (Supp Table 7).
- “T2D (RGC)” (30K cases, 350K controls), a custom trait for type-2 diabetes that we derived based on multiple available sources of data, including self-reported information (such as the phenotype above), hospital and death records (ICD10 codes). This trait was less strongly associated with MAP3K15 (OR=0.85, $P=2.8 \times 10^{-6}$) but had the strongest genetic correlation with glycated haemoglobin A1c. For this reason, it is the phenotype that is flagged in the section that describes protective associations with diseases that are genetically correlated with a quantitative trait (Supp Table 12). We have added a footnote to both Supp Tables 7 and 12 to clarify this point.

Second, there was a typo in the P-value shown in line 219, which should read $P=2.8 \times 10^{-6}$ (as shown in Supp Table 12) and not $P=2.3 \times 10^{-6}$. This has now been fixed, together with the typo in the gene name found in the abstract.

7. In the cross-ancestry analyses, the authors report that "Similar results were observed for binary traits (fig. S9b)" (line 243) but the concordance in Fig. S9b appears to be somewhat lower than for the quantitative traits.

We have provided more detail on this section to address this question, as also requested by reviewer #1 (comment #6). We have also updated numbers to reflect the non-redundant set of results, as per comment #3 above. Specifically, this section now reads:

"When we focused on the 564 non-redundant associations (i.e. strongest association per gene, 484 with a quantitative trait, 80 with a binary trait; Supplementary Table 6), we found that a large fraction of associations was shared across ancestries for quantitative traits but less so for binary traits, likely due to low power. Specifically, for quantitative traits, effect sizes were directionally concordant for 83% of associations in individuals of SAS, 73% of AFR and 74% of EAS ancestry, increasing to >90% when considering associations with a $P \leq 0.05$ (Extended Data Figure 7a). For binary traits, consistent effects were observed for 61% of associations in SAS, 61% in AFR and 64% in EAS (Extended Data Figure 7b). Similar results were observed when assessing directional consistency across the full set of 8,865 associations (Supplementary Data 2)."

8. The section describing the impact of burden test composition on the yield of genetic associations is interesting. The authors observe that aggregating variants across a wider range of allele frequencies generally appears more powerful (p. 13). However, are an increasing fraction of the associations potentially explained by LD with common variants as the MAF threshold rises? For example, what fraction of the 971 - 804 = 167 burden associations discovered only at $MAF < 1\%$ lost significance upon conditioning on GWAS sentinel variants? In a similar spirit, are the 804 of 1,775 associations discovered exclusively by aggregating pLOF and deleterious missense variants enriched for associations that might tag common variants?

This is an important point that was not considered in our initial analysis. Upon further investigation, we found that the reviewer's intuition is correct. Specifically, we found that:

- When considering a burden of pLOF variants, ~10% of burden tests that included variants with a MAF up to 1% were no longer experiment-wide significant after conditioning on GWAS sentinel variants (ie. P-value dropped below 2.18×10^{-11}). This contrasts with 4% when considering burden tests with variants that had a MAF up to 0.001% (see table below).

Marker type	Allele frequency bin	Associations significant after conditioning on GWAS signals?				Proportion of associations that remain significant after conditioning on GWAS signals (a/b)
		NA	No	Yes (a)	Total (b)	
Burden of pLOF variants	Singletons	0	7	192	199	0.965
	$\leq 1E-05$	7	13	493	513	0.961
	$\leq 1E-04$	12	22	746	780	0.956
	≤ 0.001	15	41	802	858	0.935
	≤ 0.01	14	86	871	971	0.897
Burden of pLOF and deleterious missense variants	Singletons	0	12	182	194	0.938
	$\leq 1E-05$	6	12	503	521	0.965
	$\leq 1E-04$	10	28	851	889	0.957
	≤ 0.001	10	70	980	1060	0.925
	≤ 0.01	8	227	1229	1464	0.839
Individual pLOF variants	$\leq 1E-05$	0	0	28	28	1.000
	1E-05 to 1E-04	1	10	172	183	0.940
	1E-04 to 0.001	3	19	103	125	0.824
	0.001 to 0.01	0	46	99	145	0.683
Individual deleterious missense variants	$\leq 1E-05$	0	3	37	40	0.925
	1E-05 to 1E-04	0	12	187	199	0.940
	1E-04 to 0.001	0	31	244	275	0.887
	0.001 to 0.01	0	157	264	421	0.627
Total		86	796	7983	8865	0.901

- This pattern was even more pronounced when we focused on associations that were exclusively found by burden associations with MAF<1% (but not at lower MAF) or burden associations that included pLOFs + deleterious missense variants (versus those that included only pLOFs), as suggested by the reviewer.

These results show that LD with common variant signals contributes to (but it's not the sole explanation for) the larger yield of genetic associations observed with burden tests that included (i) a wider range of allele frequencies; or (ii) a broader class of variants... as predicted by the reviewer.

To avoid this concern and keep this section simple and easy to interpret, we now present results (page 14, Supp Table 16, Extended Data Figure 8) using only associations that remain after conditioning on the common variant signals. In this conditional analysis, the conclusions in our original submission remain unchanged, although the counts of genes in each category change slightly. We have also included the table above as new Supp Table 17 and a summary of these findings in page 16.

9. I couldn't seem to find a row for the association between ANGPTL3 and Vitamin D (highlighted on p. 16) in Data S2. Did this signal survive conditioning on GWAS sentinel variants?

Great catch. The explanation for this corner case is the following. Supp Data 2 lists all 8,865 associations discovered at $P < 2.18 \times 10^{-11}$ prior to conditioning on GWAS sentinel variants, of which 7,990 remained significant after conditioning on GWAS signals. The association between ANGPTL3 and vitamin D is not shown in Supp Data 2 because the association P-value before conditioning on GWAS signals was $P = 3.6 \times 10^{-11}$, and so just missed our experiment-wide significance threshold. However, after conditioning on GWAS signals, the significance improved to $P = 1.2 \times 10^{-11}$. The analysis that identifies likely effector genes at GWAS loci used association results after conditioning on GWAS signals, and so ANGPTL3 is flagged in that analysis.

As other readers may notice this discrepancy, we have added a note to Supp Table 17 to explain why the association with ANGPTL3 (and an extra 13 genes) is not found in Supp Data 2.

10. I am unsure whether the HAL eQTL should be considered a "gain-of-function" variant (line 359) if it only alters expression.

We've replaced "gain-of-function" with "expression-increasing", and so this sentence now reads:

"Altogether, these results implicate HAL in both vitamin D levels and skin cancer and highlight an allelic series that includes rare loss-of-function protein-altering variants (trait-increasing) as well as common expression-increasing non-coding variants (trait-lowering)."

11. The imputation results (Fig. 4) are probably somewhat over-optimistic in that they represent the ideal scenario of imputation using a reference panel with perfectly-matched ancestry: i.e., imputation into another UK cohort would achieve the indicated accuracy, but imputation accuracy even into European-ancestry cohorts outside the UK would presumably be lower. More generalizable results could be obtained by imputing into an external data set such as 1000 Genomes (aggregating variants within MAF ranges to obtain sufficient sample sizes).

To address this point we created a single reference panel of 300,000 individuals with PCA-derived European ancestry and who self-reported as “White British”, and a separate test dataset of 49,926 individuals with PCA-derived European ancestry who did not self-identify as “White British”. This testing scenario is denoted by rows “300,000 WB” in Supp Table 22, and is directly comparable to the more closely ancestry matched scenario denoted “300,000”. As expected, the results show that imputation performance is not as good when ancestry is less well matched and we now reference these results in the main text.

In addition, and in response to a related question from reviewer #3, we measured imputation accuracy in ancestry specific subsets of the 50,000 target dataset for the 400,00 reference panel results, and have added these as extra rows to Supp Table 22. The three main subsets we used were White British, non-White British European and non-European. We also split the non-European subset down into South Asian, East Asian and African subsets. These results also show that the match in ancestry between reference and target samples impacts performance.

While carrying out these new experiments we uncovered a few small errors in how SNPs had been coded when calculating imputation r^2 , which have now been fixed and lead to some small changes to the numbers in Supp Table 22. At the same time we did some more experimentation with different models to extrapolate to larger reference panel sizes. There is no theoretical correct model to fit here, and we found some sensitivity to the details of the models we tried. For this reason, we have decided to de-emphasize these extrapolations and have moved them to Extended Data Figure 10 and Supp Figure 5. We also provide a fuller description of this analysis in the Methods section.

Referee #3

1. Replication of RV associations in African, South Asian and East Asian ancestries: please state clearly how many of the 8,865 associations identified in the European cohort replicate with a direction-consistent effect AND $P < 0.05$ in each ancestral group.

Of the 8,865 associations identified in the European ancestry group:

- 5,344 were tested in the South Asian ancestry group (N=10K), of which 1,309 (24.5%) had a $P < 0.05$ and consistent direction of effect.
- 5,174 were tested in the African ancestry group (N=9K), of which 947 (18.3%) had a $P < 0.05$ and consistent direction of effect.
- 2,764 were tested in the East Asian ancestry group (N=2K), of which 428 (15.5%) had a $P < 0.05$ and consistent direction of effect.

However, because the sample size for each of these groups is very small, power was inadequate to use these data for formal replication (as we did in the DiscovEHR cohort). This is why we instead focused only on comparing effect sizes between European and non-European ancestries. Nonetheless, we have added summary statistics from each of these ancestral groups to Supp Data 2, so that it's very easy for readers to obtain this information.

Related to this, reviewer 2 asked us to try to focus the main text on a set of non-redundant associations (see comment #2 above). We agree this is important to facilitate interpretation of results for some analyses, such as when we assessed replication in the DiscovEHR cohort, and for this lookup of European associations in non-European ancestries. As such, we updated this section to focus on the strongest trait-variant association per gene, so 564 associations in total (those presented in Supp Table 6), instead of 8,865. Results are very similar, with 89 of 386 (23.1%) associations tested replicating in South Asian, 77/378 (20.4%) in African, and 38/221 (17.2%) in East Asian ancestries. Supp Data S2 includes a column that can be used to filter down to summary statistics for the 564 lead associations. We've also updated Extended Data Figure 7 to show results for the non-redundant set of 564 associations and, for completeness, we kept the original comparison of effect sizes (which were based on the full set of 8,865 associations) in Supp Figure 6.

2. Replication of RV associations in the Geisinger DiscovEHR cohort: the current presentation of these results is confusing and it is not clear why only the 279 "strongest trait-variant associations" are being tested for replication. Please state clearly how many of the 8,865 associations identified in the UKBB cohort have a matching phenotype in the Geisinger cohort, and how many of these associations replicate with a direction-consistent effect AND $P < 0.05$ (similar to #2 above).

The reason we focused the replication analysis on a single association per gene (so 564 associations in total) was to remove redundancy among the 8,865 associations due to having tested multiple variants per gene and multiple traits per variant. For example, 344 of the 8,865 associations were with *TTN*, 220 with *MC4R*, 182 with *TET2*, etc. Instead of attempting to replicate every association per gene, we only attempted to replicate the most significant association per gene. This maximized power, facilitated interpretation of results (as associations are mostly uncorrelated) and is in line with comment #2 from reviewer 2. We have clarified this point in the main text. Of the 564 associations, 279 were available for replication, of which 193 (69%) had a $P < 0.05$ and the same direction of effect.

However, it is likely that some readers may be interested in knowing if associations other than the strongest trait-variant pair per gene replicate in the DiscovEHR. Therefore, we have expanded the lookup in DiscovEHR to all 8,865 associations, and added the results to Supp Data 2. Of the 8,865 associations, 4,083 were available in the DiscovEHR cohort, of which 2,855 (70%) had a $P < 0.05$ and the same direction of effect. This information has been added to page 13.

3. Phenotype definitions and association analyses: please provide more information on the frequency distribution of the tested binary phenotypes; how many phenotypes can be considered as rare diseases vs. more common traits? Also, please clarify how sex chromosomes and sex-specific phenotypes (e.g. prostate cancer, breast cancer etc.) are being handled in the analysis.

Great suggestion, thank you. The frequency distribution of the binary phenotypes tested is shown below in graphical and tabular form. We tested 3,706 binary phenotypes with at least 100 cases, of which 1,875 phenotypes had a prevalence $< 1\%$, 1,136 between 1% and 10%, and 695 $> 10\%$. Prevalence was estimated by dividing the number of cases by the total number of individuals with non-missing data. For female-specific phenotypes (eg. breast cancer) males were set to missing, and conversely for male-specific phenotypes. We have added this information to the new Supp Figure 2 and updated the text accordingly on page 4.

We tested associations with genes on the X but not Y chromosomes. For the non-pseudo autosomal regions of the X chromosome, we used a dosage compensation model, with homozygous reference males coded 0, and hemizygous males coded 2. We have added this information to the Methods section.

Prevalence	Number of phenotypes
<0.1%	2
0.1% to 0.5%	1354
0.5% to 1%	519
1% to 5%	869
5% to 10%	267
10% to 20%	214
>20%	481

4. *Overlap between CV signals from GWAS and RV signals: please provide more detail on the approaches used for GWAS analyses including methods used for association testing (was REGENIE used for GWAS?), PCA adjustments (same as for RV analyses?), genomic inflation factors (not provided). Also, the $P < 10^{-7}$ to define independently associated signals by GCTA-COJO for inclusion in the conditional analyses of RV associations may need a better justification -- how was this threshold selected?*

Yes, REGENIE was used for the analysis of both exome sequencing and imputed data. We described this in the Methods section (under “Genetic association analyses”, page 36), including a description of PC adjustments. We have added additional text to this section to clarify other elements of the analysis, including the analysis of the X chromosome (as requested above).

We used a $P \leq 10^{-7}$ to ensure that we included in the subsequent conditional analyses of exome sequencing data any common variant signals that were close to (but not quite surpassed) the more commonly used genome-wide significance threshold of $P < 5 \times 10^{-8}$. At $P < 10^{-7}$ we identified 114,389 independent GWAS signals, whereas at $P < 5 \times 10^{-8}$ we identified 7,114 fewer signals (107,275 in total). This clarification has now been added to the Methods section (page 42).

5. Beyond 500,000 exomes: Please clarify in the results if the projections of pLOF carriers for a dataset of 5 million sequenced individuals are based on the UKBB Europeans, or all ancestries included in the UKBB. Given that UKBB is dominated by Europeans, would similar projections be expected for sequencing studies of more diverse populations?

The projections of pLOF carriers for a dataset of 5 million sequenced individuals are based on all ancestries included in the UK Biobank study which, as the reviewer points out, is disproportionately (95%) European. How does this affect our results? Specifically, had we sequenced a more ancestrally diverse cohort, would our projections for 5 million individuals be very different? We performed new analyses that show that this is not the case. Specifically, we predicted the number of pLOF carriers expected in 5M individuals based on (i) 46K individuals of European ancestry from the UKB; and (ii) 46K individuals from the UKB, including 23K of European ancestry and all 23K individuals of non-European ancestry (10K of South Asian, 9K of African, 2K of East Asian ancestry and 2k of admixed ancestry). As illustrated below, projections based on the more diverse set of samples (dashed line) are slightly higher but overall similar to the estimates obtained from the European-only dataset (solid line).

This figure is now included in Extended Data Figure 9c, and the main text was updated accordingly (page 21).

6. Whole-genome sequencing and imputation: The analysis of minimal imputable MAF by varying size of the reference panel is very helpful, but again this analysis is based predominantly on UKBB Europeans. The authors should comment on the relevance of these findings to the imputation of non-European genomes.

To address this point we measured imputation accuracy in ancestry-specific subsets of the 50,000 target dataset for the 400,00 reference panel results, and have added these as extra rows to Supp Table 22. The three main subsets we used were White British, non White British European and non-European. We also split the non-European subset down into South Asian, East Asian and African subsets. These results show that the match in ancestry between reference and target samples impacts performance and we now reference these results in the main text.

Please also our response to a related point from reviewer #2 who asked about European ancestry not from the UK, for which we added an additional experiment to specifically look at this point.

While carrying out these new experiments we uncovered a few small errors in how SNPs had been coded when calculating imputation r^2 , which have now been fixed and lead to some small changes to the numbers in Supp Table 22. At the same time we did some more experimentation with different models to extrapolate to larger reference panel sizes. There is no theoretical correct model to fit here, and we found some sensitivity to the details of the models we tried. For this reason, we have decided to de-emphasize the predictions and have moved them to Extended Data Figure 10 and Supp Figure 5. We also provide a fuller description of this analysis in the Methods section.

7. Results on Page 16 line 343: "Rare pLOFs and deleterious missense variants in ANGPTL3 ... were associated with higher vitamin D levels, suggesting a role for ANGPTL3 in vitamin D production in the kidney." Vitamin D is not produced in the kidney, but rather activated by 1-OH hydroxylation. As far as I know, the conclusion that ANGPTL3 would be involved in the kidney activation of vitamin D is not based on evidence - in fact, liver is a much more plausible causal tissue given that 25-OH vitamin D is the major circulating form rather than 1,25-(OH)₂. Notably, 25-OH hydroxylation of vitamin D occurs in the liver and liver is the predominant site of ANGPTL3 expression. Additionally, on page 16 line 346: "The second gene that we implicate for the first time in vitamin D biology was HAL..." For clarity, it would be helpful to mention in this paragraph that the first step of vitamin D synthesis takes place in the skin and involves UV light.

Thank you for pointing out this inaccuracy. We speculated that ANGPTL3 had a role in vitamin production in the kidney because (i) we (incorrectly) thought that vitamin D levels measured in the UKB corresponded to the active form 1,25-(OH)₂; (ii) as pointed out, activation of 25-OH vitamin D into 1,25-(OH)₂ takes place in the kidney; and (iii) kidney is the tissue with second highest expression of ANGPTL7, behind liver. But we have confirmed that vitamin D levels instead correspond to 25-OH vitamin D, which results from 25-hydroxylation by CYP2R1 in the liver. Therefore, we fully agree that liver is the plausible causal tissue underlying the association between ANGPTL3 and vitamin D levels, and have updated the relevant sentence accordingly.

We have also updated the paragraph on *HAL* as suggested, which now reads:

“The second gene that we implicate for the first time in vitamin D biology was *HAL*. The first step of vitamin D synthesis occurs in the skin and requires ultraviolet (UV) light. *HAL* is likely to play a role in this step because it encodes an enzyme that converts histidine (an essential amino acid that is incorporated into filaggrin, among other functions) into trans-urocanic acid, a major UV-absorbing chromophore that accumulates in the stratum corneum (29).”

8. The discussion section should refer directly to study limitations, which are presently buried in the supplementary text. The following should also be considered/added to the limitations: 1) defining UKBB phenotypes based on electronic health records/ICD codes or self-report may result in phenotype misclassification and/or missing data limiting the power to detect associations for some phenotypes; 2) there is still limited power to detect associations for rare phenotypes -- it would be helpful to get a better idea how many of the analyzed phenotypes are relatively rare (e.g. <0.01%, <0.1%, <1% frequency) vs. relatively common (e.g. >5%, >10%, >20% frequency) and comment on the power implications; 3) under-representation of non-European participants in the UKBB may limit the accuracy of projections for genome imputations and/or expected RV counts when expanding sequencing to more diverse populations. It would also be helpful to state as one of the limitations that the assessment of variant segregation or de novo occurrence was not possible because of the population-based design of the UKBB.

We have added a summary of limitations to the discussion as requested, and expanded the list of caveats to include the excellent points listed above.

Figure 1. please provide a more detailed legend for the HSPG2 and alkaline phosphatase example (panel C). Are the presented single variant statistics (grey) for RVs only or for RVs+CVs? I assume RVs, but please state clearly in the legend. Also the presentation of burden p-values as dots is confusing, would specify that green indicates a burden test P-value for improved clarity.

Due to space constraints, we moved panel C to a new Supp Figure 4. To address the reviewer’s question, we have also:

- 1) Updated the original locuszoom plots to only show rare pLOFs, deleterious missense and burden tests, the class of variants described in the manuscript. The original locuszoom plots had also included association results for synonymous and benign deleterious missense variants, as well as array variants (because the pgens used in the REGENIE

analysis must include array variants for step 1). This was not appropriate and so these variants have now been removed from the locuszoom plots.

2) Explained in the legend that green indicates results from burden tests; and

3) Added an extra panel to the new figure, to explicitly show results from common variants from GWAS.

Reviewer Reports on the First Revision:

Referee #1 (Remarks to the Author):

Authors carefully addressed the reviewer's comments. While several comments have not been addressed (e.g., application of two-sided rare variant test), given that the data resource will be released, scientific community will surely assess these questions in near future. Congratulation for wonderful work and great contribution to the community!

Referee #2 (Remarks to the Author):

The authors have done an impressively thorough job revising their manuscript in a short amount of time, and their revisions (which have strengthened the manuscript) satisfactorily address my previous comments. I have only a few minor suggestions the authors can consider.

1. The data on LDSC attenuation ratios and intercepts from the TOPMed-imputed GWAS is helpful; thank you for adding these columns to Supp Data 1. I was a bit surprised that the attenuation ratios for quantitative traits were somewhat higher than I have seen in the past (e.g., 0.1854 for standing height, whereas usually <0.1 indicates good control of sample structure). I am not broadly concerned about the overall results of the paper given the strong replication across cohorts and ancestries, but the authors might wish to note this caveat or investigate further.

2. I understand the need to strike a balance between presenting all results vs. non-redundant results while not confusing the reader, but I still wonder whether reporting "8,865 associations" in the abstract and several times in the main text makes sense given that a nontrivial fraction of these associations are literally completely redundant -- exactly the same burden "variant" was called multiple different names and then counted multiple times in the 8,865. The approach of retaining the most significant association per gene per trait (2,283 associations) seems like a more natural quantification of the number of results detected, but this number is not mentioned in the text.

3. I was initially confused that the allele counts for MAP3K15 contained almost as many homozygotes as hets. I eventually realized that the reason is that MAP3K15 is on the X chromosome. This might be worth noting in ST7, ST9, etc. (and changing chromosome "23" to "X" throughout would help).

Referee #3 (Remarks to the Author):

The authors have adequately addressed my comments.

Author Rebuttals to First Revision:

Referee #2

The authors have done an impressively thorough job revising their manuscript in a short amount of time, and their revisions (which have strengthened the manuscript) satisfactorily address my previous comments. I have only a few minor suggestions the authors can consider.

1. The data on LDSC attenuation ratios and intercepts from the TOPMed-imputed GWAS is helpful; thank you for adding these columns to Supp Data 1. I was a bit surprised that the attenuation ratios for quantitative traits were somewhat higher than I have seen in the past (e.g., 0.1854 for standing height, whereas usually <0.1 indicates good control of sample structure). I am not broadly concerned about the overall results of the paper given the strong replication across cohorts and ancestries, but the authors might wish to note this caveat or investigate further.

We have investigated this further and found that the relatively high attenuation ratios we observed for quantitative traits were due to the version of LDSC that we used, which implements what's referred to as the LDSC "original model" (PMID 25642630). As noted previously by others (Loh et al. 2018, Nat Genet, PMID 29892013; see their Supp Tables 4 and 5), attenuation ratios estimated with the LDSC "original model" for quantitative traits in the UK Biobank are often >0.1 , which is not the case when estimated with the more recent LDSC "baseline model" (PMID 28892061). Consistent with this, we re-estimated the attenuation ratios for all 492 traits with the LDSC "baseline model" and found that they were consistently <0.1 . For example, for height, we had obtained an attenuation ratio of 0.1851 with the LDSC "original model", whereas with the LDSC "baseline model" we get 0.0967 (0.1085 if we re-use the munge sumstats input file used with the "original model"). Therefore, we conclude that the relatively high attenuation ratios observed for the GWAS of quantitative traits were not due to unmodelled population substructure in our results but instead reflected a previously described behavior of the LDSC "original model". As such, we have updated Supp Data 1 and Supp Fig 4 to now include the intercept and attenuation ratio from the LDSC "baseline model".

2. I understand the need to strike a balance between presenting all results vs. non-redundant results while not confusing the reader, but I still wonder whether reporting "8,865 associations" in the abstract and several times in the main text makes sense given that a nontrivial fraction of these associations are literally completely redundant -- exactly the same burden "variant" was called multiple different names and then counted multiple times in the 8,865. The approach of retaining the most significant association per gene per trait (2,283 associations) seems like a more natural quantification of the number of results detected, but this number is not mentioned in the text.

Agree, and have removed "8,865 associations" from the abstract and now explicitly state on page 5 that the non-redundant set of 2,283 associations can be obtained from Supp Data 2. In the previous revision we focused most sections of the manuscript on the even stricter set of 564 non-redundant associations (eg. associations in non-European ancestries, replication in DiscovEHR). The only remaining sections that focus on the full set of 8,865 associations are "Impact of burden test composition" and "Enrichment of associations in GWAS loci", as we believe this is more appropriate.

3. I was initially confused that the allele counts for MAP3K15 contained almost as many homozygotes as hets. I eventually realized that the reason is that MAP3K15 is on the X chromosome. This might be worth noting in ST7, ST9, etc. (and changing chromosome "23" to "X" throughout would help).

Done as requested.